# GNSS signal-based snow water equivalent determination for different snowpack conditions along a steep elevation gradient

Achille Capelli[1,2], Franziska Koch[3], Patrick Henkel[4], Markus Lamm[4], Florian Appel[5], Christoph Marty[1], Jürg Schweizer[1]

[1] WSL Institute for Snow and Avalanche Research SLF, Davos, Switzerland
[2] Geophysical Institute, University of Alaska, Fairbanks, USA
[3] Institute for Hydrology and Water Management, BOKU University of Natural Resources and Life Sciences, Vienna, Austria
[4] ANavS GmbH, Munich, Germany
[5] VISTA Remote Sensing in Geosciences GmbH, Munich, Germany

*Correspondence to*: Achille Capelli (acapelli@alaska.edu) and Franziska Koch (franziska.koch@boku.ac.at)

**Abstract.** Snow water equivalent (SWE) can be measured using low-cost Global Navigation Satellite System (GNSS) sensors with one antenna placed below the snowpack and another one serving as a reference above the snow. The underlying GNSS signal-based algorithm for SWE determination for dry- and wet-snow conditions processes the carrier phases and signal strengths and derives additionally liquid water content (LWC) and snow depth (HS). So far, the algorithm was tested intensively for high-alpine conditions with distinct seasonal accumulation and ablation phases. In general, snow occurrence, snow amount, snow density and LWC can vary considerably with climatic conditions and elevation. Regarding alpine regions, lower elevations mean generally earlier and faster melting, more rain-on-snow events and shallower snowpack. Therefore, we assessed the applicability of the GNSS-based SWE measurement at four stations along a steep elevation gradient (820, 1185, 1510 and 2540 m a.s.l.) in the eastern Swiss Alps during two winter seasons (2018-2020). Reference data of SWE, LWC and HS were collected manually and with additional automated sensors at all locations. The GNSS-derived SWE estimates agreed very well with manual reference measurements along the elevation gradient and the accuracy (RMSE = 34 mm, RMSRE = 11 %) was similar under wet- and dry-snow conditions, although significant differences in snow density and meteorological conditions existed between the locations. The GNSS-derived SWE was more accurate than measured with other automated SWE sensors. However, with the current version of the GNSS algorithm, the determination of daily changes of SWE was found to be less suitable compared to manual measurements or pluviometer recordings and needs further refinement. The values of the GNSS-derived LWC were robust and within the precision of the manual and radar measurements. The additionally derived HS correlated well with the validation data. We conclude that SWE can reliably be determined using low-cost GNSS-sensors under a broad range of climatic conditions and LWC and HS are valuable add-ons.

# 1    Introduction

The water stored in the seasonal snow cover plays a crucial role in the hydrological cycle in mountain regions and is a key source of fresh water supply. The snow water equivalent (SWE) expresses the amount of water stored in the snow, which together with its melt rate influences river runoff with large effects on agriculture, hydropower production, water supply and

ecosystems downstream of mountain head-watersheds and can contribute to floods, slush flows and other natural hazards. Estimating SWE in high temporal resolution as well as its spatial distribution is a major task in snow hydrology (Dozier et al., 2016; Largeron et al., 2020). On the other hand, snow effects the climate system due to its high reflectivity, insulation properties and cooling effects and is, therefore, an essential climate variable (Bojinski et al., 2014). Monitoring the temporal and spatial distribution of the snow mass is hence essential for assessing the water storage in snow and subsequent runoff for climatological

applications (e.g. Marty et al., 2017). Moreover, measuring SWE is necessary for the development of building codes and monitoring current snow loads to guarantee the stability of structures.

Despite the need for monitoring SWE for various applications, and although different methods exist for estimating SWE, encompassing in situ measurements, remote sensing and physically-based modelling, as well as combinations thereof and assimilation techniques, continuous measurements are often not available or feasible, especially in complex topography such

as mountain areas. For large subarctic areas, the spatial and temporal distribution of SWE under dry-snow and rather shallow snowpack conditions can be obtained from microwave satellite remote sensing (Larue et al., 2017; Pulliainen and Hallikainen, 2001; Shi and Dozier, 2000). This is, however, until now, not sufficiently feasible in highly complex alpine terrain due to either low spatial resolution especially of passive microwave sensors or regarding active microwave sensors, due to penetration depth limits, foreshortening, shadowing and layover effects. However, recent developments including Sentinel-1 radar

observations seem promising (Lievens et al., 2019; Lievens et al., 2021; Marin et al., 2020; Tsang et al., 2021). In contrast to SWE, snow depth (HS) can accurately be determined with various methods even for alpine catchments. This encompasses the application of satellite stereo images (Deschamps-Berger et al., 2020), airborne LiDAR altimetry approaches (Deems et al., 2013; Helfricht et al., 2014), photogrammetric reconstructions, using images taken by drones (Avanzi et al., 2018; Bühler et al., 2017), or terrestrial LiDAR surveys (Grünewald et al., 2010; Prokop et al., 2008). However, for the conversion of the HS

products into SWE additional density information, e.g. using modelling approaches (Jonas et al., 2009; Winkler et al., 2021) or additional measurements, is still needed (Dozier et al., 2016), which is not at every location available or easy to obtain. SWE can also be derived by physically-based modelling (e.g. Le Roux et al., 2020; Lehning et al., 2006). However, the results depend largely on the quality and availability of meteorological input data and should be validated against in situ measurements. The best results in distributed modelling at high resolution (250 m) (Griessinger et al., 2019) are achieved by

assimilating either space born or in situ observations (Magnusson et al., 2017; Winstral et al., 2019).

Hence, point measurements of SWE are still essential, for data assimilation, validation and calibration of models and remote sensing data. Moreover, long-term time series of SWE measurements, which only few exist, are particularly valuable for climate change monitoring (Marty et al., 2017; Mote et al., 2018). Traditionally, SWE is measured by weighting a given

volume of snow (Haberkorn, 2019). At a limited number of stations worldwide such manual measurements are performed
(bi-)weekly. Manual SWE measurements are, however, non-continuous, time costly, destructive and often sparse, especially
in remote and mountainous terrain. Common approaches providing continuous data are gravimetric sensors, sensors based on
natural gamma radiation and cosmic ray sensors (Haberkorn, 2019; Pirazzini et al., 2018). Gravimetric sensors such as snow
pillow and snow scale, which measure SWE by weighing the overlaying snow cover, are costly, difficult to install and prone
to errors due to bridging effects in the snow cover, non-natural heat-flux and drainage effects (Johnson and Schaefer, 2002;
Johnson et al., 2015). Passive gamma radiation instruments determine SWE from the attenuation of the natural gamma
radiation emitted and travelling through the snow, but can only measure SWE < 600 mm with reasonable accuracy (Haberkorn,
2019). In recent years, cosmic ray sensors showed good results in deriving SWE from the absorption of natural fast neutrons
in the snow cover and the consequent attenuation of the neutron count (Gugerli et al., 2019; Schattan et al., 2017; Schattan et
al., 2019). A comparison of the performance of different radiation-based field sensors for monitoring SWE can be found in
Royer et al. (2021).

In the last decade, promising approaches emerged that use L-band microwave signals transmitted from Global Navigation
Satellite System (GNSS) satellites to derive continuously and non-destructively snow cover properties. On the one side, HS
can be derived with reflectometry techniques using antennas, which are permanently installed above the ground (Botteron et
al., 2013; Jin and Najibi, 2014; Larson et al., 2009). However, to obtain SWE, some external information on snow density is
needed. On the other side, recently a GNSS method to directly derive SWE was developed using low-cost GNSS sensors
installed above and below the snow cover. SWE is derived by using a combined approach of carrier phases measurements and
signal strength information, retrieving the time delay and attenuation of the GNSS signals in the snowpack. The development
of the current algorithm with all processing steps described in Koch et al. (2019) is the result of merging several steps of
development. In a first step, Koch et al. (2014) derived the liquid water content (LWC) of a snowpack from the attenuation of
the GNSS signals travelling through the snow cover. Combining the GNSS signal attenuation approach of Koch et al. (2014)
with two-way travel time information derived by an L-band upward-looking ground penetrating radar (upGPR), it was possible
to simultaneously derive SWE, HS and LWC for dry- and wet-snow conditions (Schmid, 2015; Schmid et al., 2015). However,
radar systems are rather expensive and the data retrieval still needs manual supervision. In a further step, Henkel et al. (2018)
exploited the GNSS carrier phase measurements for deriving SWE with a low-cost GNSS system for dry-snow conditions. A
similar approach relying on carrier phase measurements allowed an hourly SWE estimation from the GNSS signal (Steiner et
al., 2018; Steiner et al., 2019a; Steiner et al., 2019b). Koch et al. (2019) generalized the techniques of Koch et al. (2014),
Schmid et al. (2015) and Henkel et al. (2018) for dry- and wet-snow conditions by combining GNSS carrier phases and signal
strength, snow permittivity models and a simple snow densification model to simultaneously derive SWE, HS and LWC with
only one GNSS sensor system.

The GNSS algorithm described by Koch et al. (2019) includes different snow densification assumptions for dry and wet snow,
allowing HS derivation for both conditions. The two density model assumptions were first developed for high-alpine seasonal
snowpack conditions (Koch et al., 2015; Koch et al., 2019; Schmid et al., 2014), which are characterized by distinct seasonal

accumulation and ablation phases. Good accuracy of the GNSS-derived SWE was achieved for the high-alpine site Weissfluhjoch (2540 m a.s.l) near Davos, Switzerland, were the algorithm was intensively tested and validated (Koch et al.,

2019). The algorithm was further tested at sites in Newfoundland and the Canadian subarctic where the accumulation phase is also clearly separated from the ablation phase and was integrated in the SnowSense® GNSS sensor system (Appel et al., 2019). In low-elevation areas of the Alps, however, the snow cover is overall shallower and the density evolution might differ considerably from the high-alpine site Weissfluhjoch due to different meteorological conditions. Also, there is often no clear separation into an accumulation period with dry-snow conditions and an ablation period with wet-snow conditions. Instead,

transitions from wet to dry snow frequently occur due to positive air temperatures and rain-on-snow events. Moreover, for shallow snowpacks the daily melt-freeze cycle in the upper layers affects the bulk snow cover properties more than for a deep snowpack.

Therefore, in this study, we aim to assess the performance of the GNSS algorithm described by Koch et al. (2019) for locations with mainly elevation-dependent differences in SWE and snow depth, frequency of changes between dry- and wet-snow

conditions, densification and influence of rain events. To this aim, we installed SnowSense® GNSS stations and performed validation measurements along a steep elevation gradient (from 820 to 2540 m a.s.l.) for two winter periods (2018-2019 and 2019-2020). While our focus is on the accuracy of the GNSS-derived SWE, we also assessed the accuracy of water equivalent of daily snowfall, LWC and HS. Finally, we discuss the advantages and limitations of an operational use of the GNSS system for SWE derivation in general and point out potential future development steps.

**2     GNSS measuring principles**

The target value of the GNSS approach is SWE, whereas HS and LWC are rather considered by-products. The GNSS algorithm applied for this study is based on differential GNSS measurements using microwave L1-band signals with a central frequency of 1.57542 GHz (wavelength ca. 19 cm) encompassing signals of the U.S. Global Positioning System (GPS) and the European Galileo system. Each GNSS-based SWE sensor consists of two GNSS receivers and antennas. One of the antennas is placed

on the bare ground and gets subsequently covered by snow. The second antenna acts as a reference and is placed above the snow cover (Figure 1), e.g. on the top of a pole. Snow on the ground has a clear impact on the GNSS carrier phase measurements received at the buried antenna and in case of wet snow, also on signal strength since signal attenuation increases with increasing LWC. Atmospheric delays from the ionosphere and troposphere as well as satellite position, clock offset, phase and code bias errors affect the measurements of both the upper and lower antenna. The differential processing of the GNSS

signals (using double difference measurements) eliminates these errors and keeps only the snow information, the relative position between the two antennas (also called baseline vector), the double differenced carrier phase integer ambiguities and the double difference measurement noise and multipath propagation (Henkel et al., 2018). In case of no snow, the relative position is determined with standard RTK positioning with millimeter-level accuracy. The relative position is then considered as a known parameter and does no longer need to be estimated during the winter season.

Under dry-snow conditions, the SWE information is included directly in the differential carrier phase measurements. More specifically, the differential carrier phase measurements are a linear function of SWE and the carrier phase integer ambiguities. The mapping between SWE and the differential carrier phase measurements depends on the elevation of the refracted satellite signals and the speed of signal propagation in dry snow, which depends on snow density; we used a mean value $v_{s,dry} = 2.3 \times 10^8$ m s$^{-1}$ as suggested by Schmid et al. (2014). The mapping between the carrier phase integer ambiguities and the differential

carrier phase measurements depends only on the signal wavelength (19 cm in L1 band) and is straightforward.

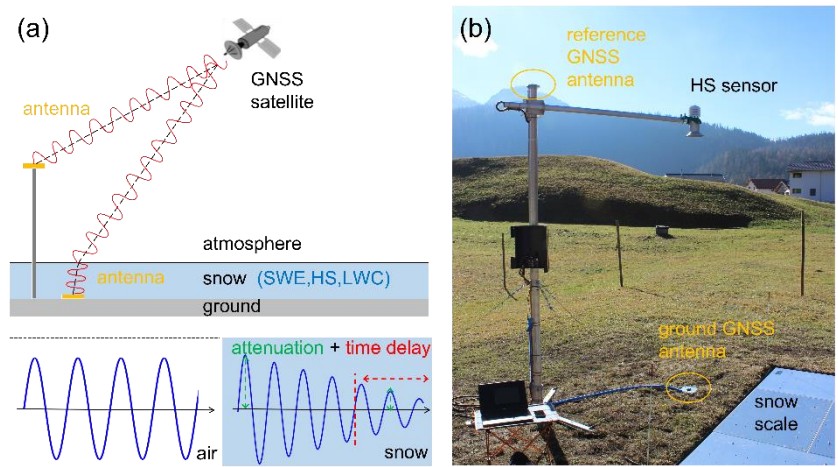

**Figure 1: a) Schematic representing the GNSS sensor setup and the measuring principles. The bottom graphics illustrate the phase delay and attenuation of the GNSS signal in snow. b) Measuring station at site Laret with GNSS sensor setup as well as automated HS and SWE sensors for validation. The pressure sensor "snow scale" is partially visible in the lower right corner.**

For the derivation of SWE under wet-snow conditions, the carrier phases processing is similar, however, as the speed of signals in wet snow $v_{s,wet}$ depends on LWC, also signal strength information has to be considered. According to Koch et al. (2014), LWC can be derived by GNSS signal strength, HS and permittivity models for wet snow. For the latter, we applied for the real part the dielectric three-phase mixing model after Roth et al. (1990) and for the imaginary part the semi-empirical equation after Tiuri et al. (1984). Therefore, in case of wet snow, a combined approach of using time delay, signal strength and an

information on HS is necessary to derive SWE, which is explained in detail in Koch et al. (2019).

In the entire combined approach, HS is considered as a supporting value and its calculation is based on simple snow densification models, which differ for dry and wet snow. In case of dry-snow conditions, HS is calculated based on the GNSS-derived SWE of the current time step as well as the SWE evolution of all previous times steps with continuous snow cover on the ground by assuming that densification follows an exponential behavior with time (Koch et al., 2019). For dry-snow

conditions, each layer has a specific density $\rho_{s,dry,t}$ at a certain time step $t$ and the layer densifies over 30 days with an exponential densification rate of 1/6 d$^{-1}$ up to a set maximum dry-snow density $\rho_{s,dry,max} = 357$ kg m$^{-3}$ as proposed for the site

Weissfluhjoch by Schmid et al. (2015). If SWE increases from the previous time step of measurements, a new snow layer with an initial density $\rho_{s,0} = 100$ kg m$^{-3}$ is added to the model and densifies over time.

For wet-snow conditions, we used the bulk densification approach described in Schmid et al. (2015) which largely depends on
the LWC. The input variables for this approach are LWC and SWE derived for the current time step and as starting value for density the defined maximum dry snow density $\rho_{s,max}$. The upper bound of snow densification is set to $\rho_{s,wet,max} = 600$ kg m$^{-3}$ HS is then derived for both wet and dry-snow conditions with $HS = \frac{SWE}{\rho_s}$ with $\rho_s$ being the bulk snow density of either dry or wet snow. So far, the implemented dry- and wet-snow density model assumptions worked well for the high-alpine seasonal snowpack evolution with distinct accumulation and ablation phases.

The main processing steps for the derivation of SWE, LWC and HS from the GNSS signals under either dry- or wet-snow conditions are summarized schematically in Figure 2. First, a distinction between dry- and wet-snow conditions is made based on a GNSS signal strength threshold. The processing in case of dry snow is straightforward, and in addition to SWE, an estimate of HS is given by applying the integrated dry-snow densification model. In contrast to dry snow, the processing of wet snow is more complex. SWE, HS, LWC, snow density and signal speed $v_s$ are derived in multiple iterative steps from
phase delay and signal attenuation starting and using the snow density and signal speed of the previous time step (e.g., previous day) as initial values. For more details on the dry- and wet-snow GNSS algorithm see Henkel et al. (2018) and Koch et al. (2019).

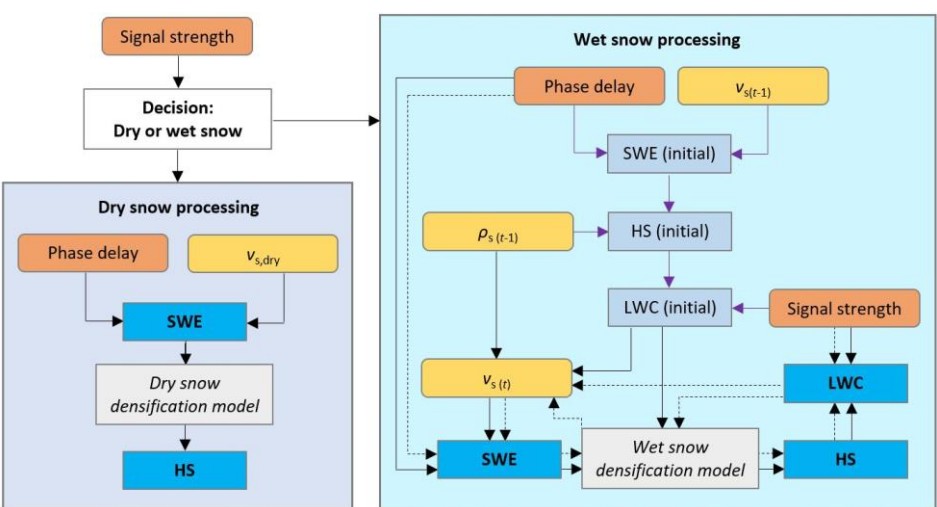

**Figure 2: Diagram illustrating the main processing steps for the derivation of the snow cover properties SWE, LWC and HS (blue boxes) using as GNSS input phase delay, derived from the differential carrier phase measurements and signal strength (orange**
**boxes). Snow density and the velocity of signals in dry and wet snow are additional inputs or intermediate variables (yellow boxes). For dry snow, the processing is straightforward and a constant value $v_{s,dry} = 2.3 \times 10^8$ m s$^{-1}$ is assumed. Regarding the wet-snow processing chain, SWE, HS and LWC are first derived as initial values (violet arrows) using snow density $\rho_s$ and the velocity $v_s$ information of the previous time step; in a second step (black arrows), two iterative calculation steps (second circulation is marked with dashed lines) follow to derive the final values.**

## 3    Study sites and data

The four sites selected for the study were the stations Küblis, Klosters, Laret and Weissfluhjoch, situated in close geographical vicinity (within a radius of 6 km) and covering a steep elevation gradient ranging from 820 to 2540 m a.s.l. in the region of Davos (eastern Swiss Alps). Table 1 provides a summary of the station characteristics. The snow cover at the study sites can be considered as representative for the respective elevation in this area. Data were collected during the winter seasons 2018-2019 and 2019-2020.

The high-alpine site Weissfluhjoch is located on a flat part of a valley at 2540 m a.s.l., which is well protected from strong wind and has a permanent snow cover for about two thirds of the year. The site is equipped with automated snow and meteorological sensors (Marty and Meister, 2012). A snow scale and a snow pillow continuously record SWE and an ultrasonic sensor snow depth. In addition, daily at 8:00, an observer measures snow depth, height of new snow (HN) and water equivalent of snowfall (HNW). Daily manual observations from a second snow depth pole ($HS_2$) in the immediate vicinity of the snow pillow and scale are available as well. LWC was measured automatically with an upward-looking ground penetrating radar (upGPR) according to Schmid et al. (2014) in the winter 2019-2020. A good overview of the location and the sensors is given in Schmid et al. (2015) or Koch et al. (2019). At this site, GNSS measurements are ongoing since several years (Henkel et al., 2018; Koch et al., 2014; Koch et al., 2019; Steiner et al., 2019b); the data presented here are new and were not previously used. The site of Laret is located on an open meadow at 1510 m a.s.l, and is wind sheltered resulting in a very uniform snow depth. The Laret site is a CryoNet station belonging to the GCW CryoNet cluster "Davos" (Wiesmann et al., 2019) and is equipped with automated snow and meteorological sensors: SWE is measured with a snow scale, HS with an ultrasonic and two laser sensors and precipitation with a heated pluviometer. The GNSS ground antenna was placed in close proximity of the snow scale and the ultrasonic snow depth sensor (< 1 m) (Figure 1b). The measuring site in Klosters is located at 1210 m a.s.l. in a private garden. In the immediate vicinity of the GNSS ground antenna we installed an automated laser snow depth sensor. In addition, an automated air temperature sensor (radiation shielded) was installed for the winter 2019-2020. Snow depth, HN and HNW (for HN > 10 cm) were measured daily by an observer. An automated and heated pluviometer is present within 200 m at the same elevation. The Küblis site is situated at 820 m a.s.l. on a lawn in front of a hydroelectric power plant. Snow depth was measured continuously by a laser sensor in the immediate vicinity of the GNSS ground antenna and an air temperature sensor (radiation shielded) was installed for the winter 2019-2020. HS, HN and precipitation (rain gauge) were manually measured each morning. For the winter 2019-2020, the plot of the manual measurements (daily data) was moved to a nearby location (distance 330 m, elevation difference 20 m). Camera pictures documenting snow conditions and snow coverage of the ground antenna are available for all sites.

Manually observed snow profiles were performed weekly for the sites of Laret, Klosters and Küblis and bi-weekly at Weissfluhjoch. The measurements included HS, SWE and snow temperature. LWC was derived from snow density (density cutter) and relative dielectric permittivity (capacitive sensor; Denoth, 1994). LWC was measured only for some of the manual profiles due to the time-consuming procedure and need for a trained observer.

The spatial variability of snow density is lower than of HS (Jonas et al., 2009). For this reason, all manual SWE values and the SWE data recorded with the snow pillow and scale at the Weissfluhjoch site were scaled to the nearby reference depth

measurement (laser or ultrasonic HS gauges) with: $\text{SWE}' = \text{SWE}\,\frac{\text{HS}_{\text{ref}}}{\text{HS}_{\text{SWE}}}$, where $\text{HS}_{\text{ref}}$ is the reference snow depth from the automated sensor and $\text{HS}_{\text{SWE}}$ is the snow depth recorded in the snow pit or at the snow depth pole ($\text{HS}_2$) near the snow pillow and scale.

At each of the four sites, we installed a SnowSense® GNSS sensor system. It consisted of two GNSS antennas and receivers, an onboard processing, a communication module (for data transfer via the mobile phone network) and a power management

unit. The integrated u-blox LEA-M8T GNSS receivers are Multi-GNSS receivers that can receive both GPS and Galileo signals (Lamm et al., 2018). GNSS receivers provide raw data with a rate of 10 Hz or even higher. However, the receiver-satellite geometry as well as the SWE is changing only at much lower rates. Therefore, we have chosen a measurement rate of only 1 Hz for the raw data. A continuous carrier phase tracking during the measurement is still essential to prevent the need for a re-estimation of the carrier phase integer ambiguities.


**Table 1: Summary of the station characteristics. An asterisk (\*) indicates that the measurements were only available during the second winter season 2019-2020; for the winter season 2018-2019 we used data from nearby stations. HN stands for height of new snow and HNW for water equivalent of snowfall.**

|  | Weissfluhjoch | Laret | Klosters | Küblis |
|---|---|---|---|---|
| Elevation (m a.s.l.) | 2540 | 1510 | 1210 | 820 |
| Coordinates | 46°49'47''N, 9°48'34''E | 46°50'2''N, 9°52'17''E | 46°51'49''N, 9°53'17''E | 46°54'48''N, 9°46'54''E |
| SWE manual | bi-weekly | weekly | weekly | weekly |
| SWE auto | pillow and scale | scale | no | no |
| HS manual | Daily | no | daily | daily |
| HS sensor | ultrasonic | ultrasonic | laser | laser |
| Pluviometer | automated, heated | automated, heated | automated, heated | manual daily |
| HN manual | Daily | no | daily | daily |
| HNW manual | Daily | no | only if, HN > 10 cm | no |
| upGPR | yes* | no | no | no |
| Temperature | Yes | yes | yes* | yes* |

The choice of the measurement duration for the determination of a set of snow parameters is mainly driven by two factors: On the one hand, the measurement period must be sufficiently long to enable a separation of SWE and carrier phase integer ambiguities. As the satellite geometry is changing only slowly over time and as satellites are visible up to 6 hours per pass, a time span of at least 6 hours is recommended as it is necessary to capture as many satellites as possible with both, ascending and decreasing tracks. On the other hand, the measurement period should not be too long to be able to account for changes in

SWE. We have chosen a 12-hour measurement period as it provides the best trade-off between accuracy and latency; it is slightly better than using a 6-hour period as the majority of satellites used for the SWE-derivation are completely tracked in ascend and descend. On the other hand, increasing the time spam (e.g., 24 h) results in a negligible improvement. If data sets

were shorter than 12 hours, we still accepted data sets longer than 6 hours but discarded shorter data sets to avoid outliers in GNSS-based SWE determination.

The data collection and processing with 12-hour measurement periods was successful for the site at Weissfluhjoch for both winter seasons and for the majority of times at the other three sites in winter 2019-2020. In winter 2018-2019, we were faced with a firmware issue at the sites in Laret, Küblis and Klosters that caused temporally shorter data sets with irregular time intervals and some data gaps of up to two days in Küblis and Klosters and one data gap of four days in Laret (April 2019). The outages could be significantly reduced in the season 2019-2020 with only very few data sets of less than 6 hours.

Unfortunately, the unusually large snowfall in mid-January 2019 caused a bending/tilting of the station masts at the sites Klosters and Küblis. The bending and/or tilting of the mast affects the relative position between the two GNSS antennas and therefore compromises the validity of the calibration process and the derivation of snow parameters. The masts were replaced and the data recording was continued. The subsequent data were evaluated in post-processing as a re-calibration could only be performed after snow melt. The tilted masts caused data gaps at the site Klosters from 14 January to 17 February 2019 and at

the site Küblis from 14 January to 4 March 2019.

The data recording at the Laret site started at 14 December 2019, one month after the beginning of the snow accumulation, due to some issues with the initial GNSS system set-up. An issue with the data logging at the Laret site resulted in a premature end of the data sets in mid-April.

A corrosion at a cable connection at one receiver at the site Klosters caused a short gap (17-25 January 2020) that could be

easily fixed by cleaning the connection. In general, GNSS-derived SWE is quite robust to such data gaps, but not HS (see also Section 6.1 in the Discussion). To have plausible HS starting values after larger data gaps during the snow-covered period, an independently measured value of HS (automated snow depth sensor) was used as input for the GNSS algorithm.

During the two winter seasons 2018-2019 and 2019-2020, snowpack characteristics significantly differed between the four sites with regard to, for instance, snow depth and snow density evolution, and temperature (Appendix A), as well as rain-on-

snow events (Appendix B). Some webcam pictures illustrating the different snow conditions at the four sites can be found in in the accompanying data at Envidat (Capelli et al., 2020).

## 4 Results

### 4.1 Snow water equivalent

The seasonal evolution of the GNSS-derived SWE and the reference data for the four measuring sites along the elevation

gradient are shown in Figure 3. It is clearly visible that the temporal occurrence and the amount of snow increases with the elevation of the sites for both winter seasons. Moreover, at the sites of higher elevation the snow-covered period starts earlier, peak SWE occurs later and the melt phase is longer. The winter season 2018-2019 was characterized by few but large snowfall events and snow mass was among the largest in the last 20 years within the study area. At the site Weissfluhjoch, peak SWE

(1313 mm) was even the highest ever measured since 1936. The 10-day sum of new snow at the beginning of January 2019 was one of the largest ever measured for this region. Due to low temperatures in January, the snow depth was above average also at the lower elevation sites Klosters and Küblis. The winter 2019-2020 was particularly mild with average snow precipitation at Laret and Weissfluhjoch, but below average snow amounts at the lower elevation sites, with frequent rain-on-snow events in Klosters and non-continuous snow cover in Küblis where the snow never lasted longer than a week. The end of the melt season in 2019-2020 was approximately one month earlier at Laret and Klosters than in the previous winter 2018-2019 and 15 days earlier at Weissfluhjoch.

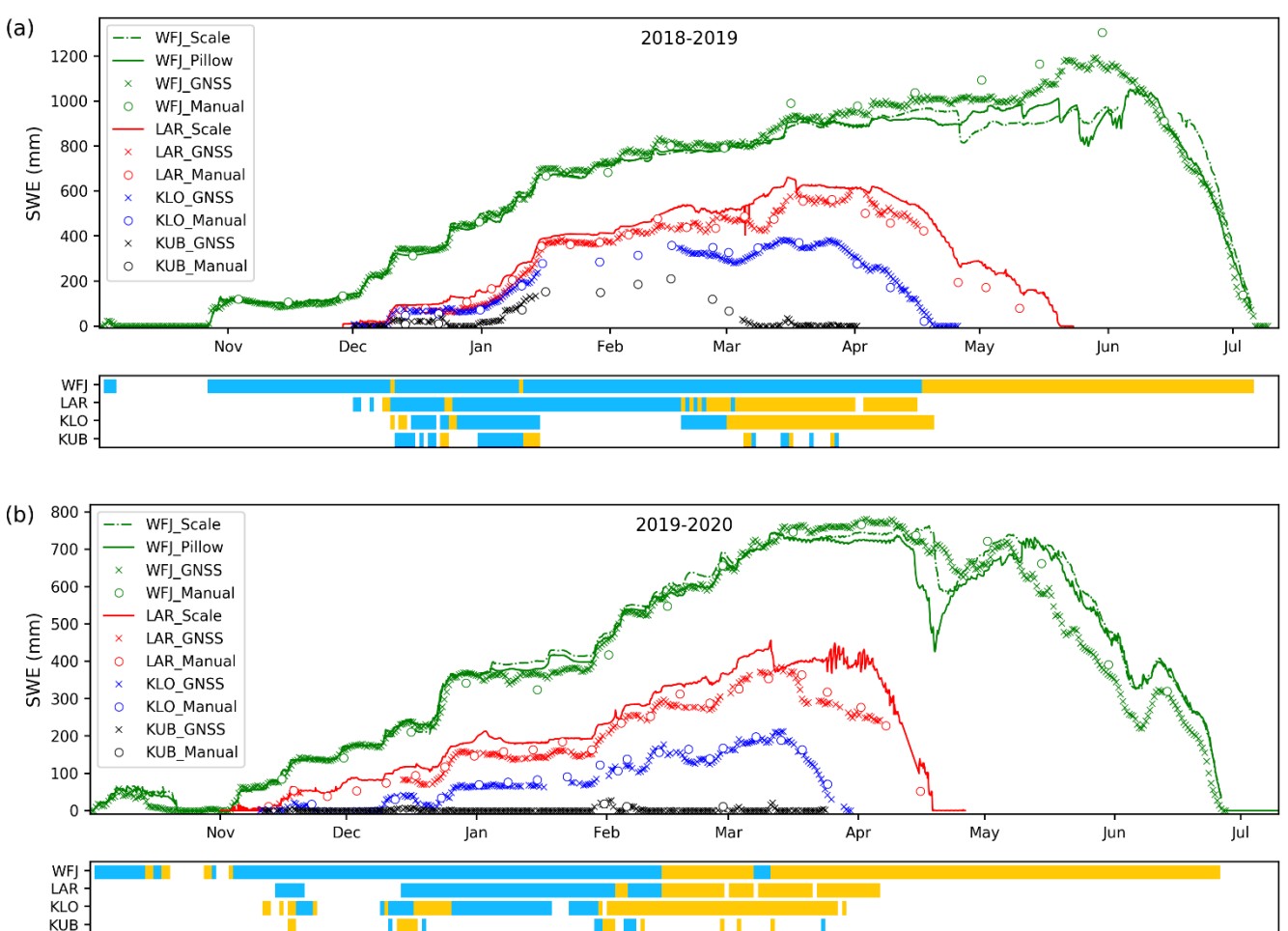

**Figure 3: GNSS-derived SWE and reference data for (a) the winter 2018-2019 and (b) 2019-2020 for the sites Weissfluhjoch 2540 m a.s.l. (WFJ), Laret 1510 m a.s.l. (LAR), Klosters 1185 m a.s.l. (KLO) and Küblis 820 m a.s.l. (KUB). The color bars indicate whether the dry-snow (blue) or wet-snow (orange) GNSS algorithm was used.**

The GNSS-derived SWE agreed well with the reference data from manual measurements, snow scale and snow pillow during both winter seasons for the three higher elevation sites. For Küblis, only a qualitative evaluation was possible as there was hardly any snow during winter 2019-2020 and a long data gap in winter 2018-2019. However, the available data from Küblis show that the GNSS system can discern very well, whether snow is covering the ground also for SWE values lower than 5 mm (see Figure C1 in Appendix C). While the GNSS-derived SWE and the manual SWE data agreed very well during the entire season, the snow scale and pillow showed some deviations at the onset of the melt period. These anomalies are visible in Figure 3 as sudden decreases in SWE (May 2019 and end of April 2020 at Weissfluhjoch, mid-March 2020 at Laret) and daily cycles (Laret, end of March 2020). For this reason, the uncertainty of the GNSS-derived SWE was evaluated relative to the manual SWE measurements. The root mean squared absolute (RMSE), relative errors (RMSRE) and $R^2$ from linear regression are shown in Table 2. Scattering and linear regression lines and parameters are shown in Figure 4.

With a very shallow snowpack the spatial variability of SWE relative to the total SWE can be very large. Accordingly, the difference between SWE above the GNSS antenna and the manually measured SWE can be large, and the relative error very high. Therefore, we considered only cases with SWE $\geq$ 25 mm in our statistical comparison. Overall, considering all sites and both winter seasons, the root mean square error (RMSE) was 34 mm and the root mean square relative error (RMSRE) was 11 %. The absolute error increased with elevation from Klosters to Laret and Weissfluhjoch and was 21, 24 and 47 mm, respectively, whereas the relative error decreased and was 15, 11 and 8 %, respectively, since at the higher elevation sites SWE was generally larger.

In addition to the entire season, we analyzed the uncertainty of the SWE measurements separately for dry-snow and wet-snow conditions; the latter ones were defined by the occurrence of liquid water (LWC > 0 %). The uncertainty of GNSS-derived SWE was also very good when dry- and wet-snow conditions were analyzed separately (Figure 4 and Table 2). In general, the absolute error was larger for wet-snow conditions, whereas the relative error was of comparable magnitude. Also, in this case the difference was mainly due to the higher amount of snow during the melt season ( $\mathrm{mean}(\mathrm{SWE}_{\mathrm{dry}}) = 300$ mm and $\mathrm{mean}(\mathrm{SWE}_{\mathrm{wet}}) = 440$ mm for all data from all sites).

Relating the SWE measured by the snow scale and pillow with the manual measurements at the sites Weissfluhjoch and Laret for two seasons and wet-snow conditions revealed that the RMSE and RMSRE were considerably higher than those obtained for the comparison with the GNSS-derived SWE. For dry-snow conditions, the errors were still higher but in general closer to the range of those for the GNSS-derived SWE. The higher uncertainty of snow scale and pillow is mainly caused by the large differences observed at the onset of the melt period for the snow pillow and scale (Figure 3).

A qualitative analysis of rain-on-snow events showed no particular influence of rain on the GNSS-based SWE estimation. See Figure B1-4 in Appendix B.

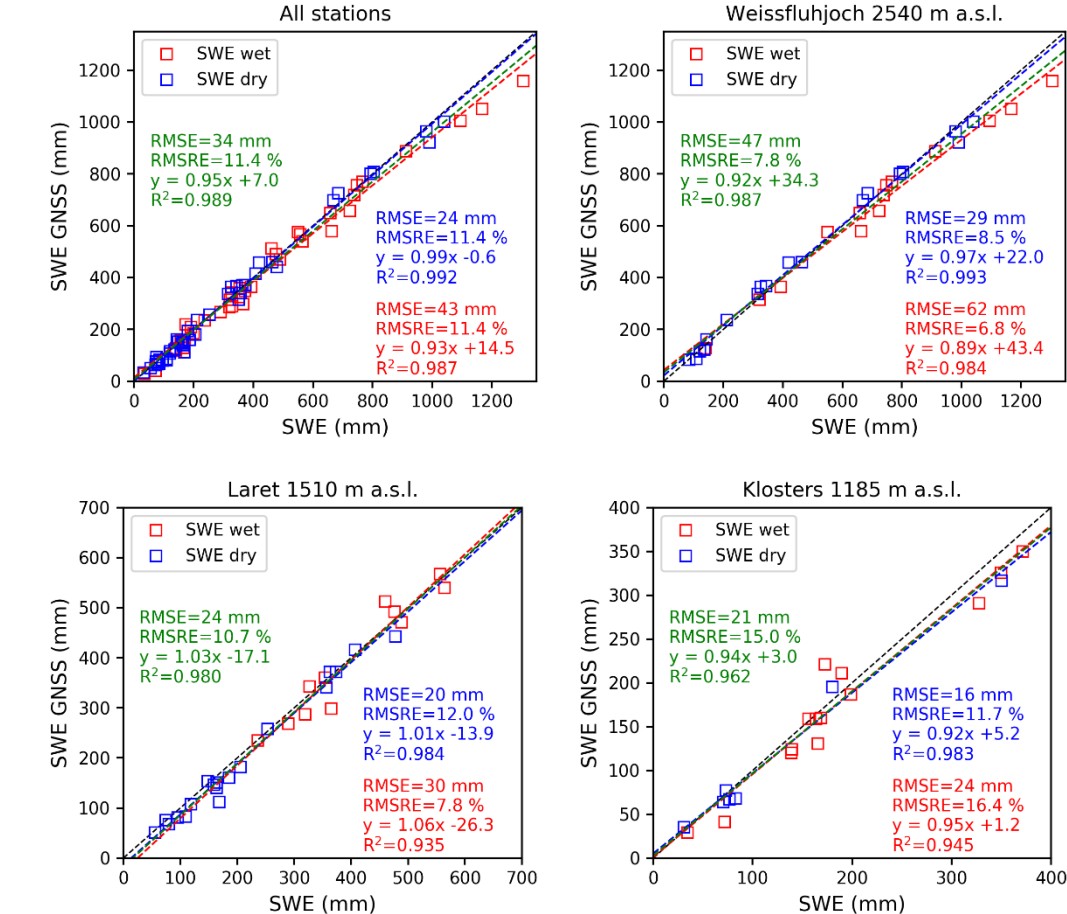

**Figure 4: Scatter plots of GNSS-derived SWE vs. manually measured SWE for dry-snow conditions (blue) and wet-snow conditions (red) for all sites together and the single sites (for both winter seasons). We do not show the data for Küblis because only few data points were available. The dashed lines represent the linear regressions (green for dry- and wet-snow data jointly). Data points with SWE < 25 mm were excluded from the analysis. The 1:1 line is shown in black.**

**Table 2: Root mean square error (RMSE), root mean square relative error (RMSRE), number of data points N and linear regression parameters (slope, intercept and $R^2$) for GNSS-derived SWE compared to the manual measurements for the single stations and all data (for both winter seasons). For the snow pillow, the data from Weissfluhjoch were used. For the snow scale, the analysis includes data from Weissfluhjoch and Laret.**

| | | RMSE (mm) | RMSRE (%) | N (-) | $R^2$ (-) |
|---|---|---|---|---|---|
| **All** | | 34 | 11 | 84 | 0.99 |
| | dry | 24 | 11 | 45 | 0.99 |
| | wet | 43 | 11 | 39 | 0.99 |
| **WFJ** | | 47 | 8 | 32 | 0.99 |
| | dry | 29 | 9 | 18 | 0.99 |
| | wet | 62 | 7 | 14 | 0.98 |
| **Laret** | | 24 | 11 | 30 | 0.98 |
| | dry | 20 | 12 | 19 | 0.98 |
| | wet | 30 | 8 | 11 | 0.94 |
| **Klosters** | | 21 | 15 | 21 | 0.96 |
| | dry | 16 | 12 | 7 | 0.98 |
| | wet | 24 | 16 | 14 | 0.95 |
| **Küblis** | | 12 | 80 | 5 | 0.95 |
| **Pillow** | | 92 | 13 | 32 | 0.95 |
| | dry | 47 | 11 | 18 | 0.99 |
| | wet | 128 | 15 | 14 | 0.91 |
| **Scale** | | 79 | 17 | 61 | 0.95 |
| | dry | 39 | 14 | 37 | 0.99 |
| | wet | 117 | 204 | 24 | 0.91 |

305

## 4.2  Detection of new snow

Some operational applications (e.g. avalanche forecasting or flood prediction) require not only an estimation of SWE of the bulk snowpack but also the daily variations indicative of snowfall and melting. Therefore, we evaluated whether the GNSS algorithm can reliably measure such variations over 24 h and 72 h by comparing these with reference precipitation data. As reference data for the Weissfluhjoch we used the water equivalent of the new snow measured manually daily at 8 a.m. For the other sites, which are less influenced by snow drift due to wind, we used the precipitation data from nearby pluviometers (automated for Klosters and Laret, and manual for Küblis). For this analysis, we used only the data from the season 2019-2020, since the GNSS-derived SWE for Laret, Küblis und Klosters for the season 2018-2019 was available only at irregular time intervals and determining the daily change in SWE (ΔSWE) was not feasible.

Snow melt results in a decrease in total SWE and consequently in negative values of $\Delta SWE_{GNSS}$ that were not measured with the reference method. Therefore, we did a separate analysis for dry-snow conditions when decreases in SWE are not expected. Figure 5 shows scatter plots of ΔSWE for 24 h and 72 h for the GNSS-derived, the snow pillow and the snow scale data versus

the reference data for all sites for winter 2019-2020. The linear regressions were computed only for days with considerable precipitation, i.e. for reference changes $\Delta SWE_{ref} > 10$ mm within 24 h or $\Delta SWE_{ref} > 20$ mm within 72 h. The GNSS-derived daily $\Delta SWE$ relative to reference data showed considerable scatter with an RMSE of 11 mm and an RMSRE of 65 % for dry-snow conditions (Fig. 5a). For the entire season (dry- and wet-snow conditions), RMSE and RMSRE were 12 mm and 72 %, respectively. Considering the 72-hour time period (Fig. 5b), the relative errors (RMSRE) were slightly smaller, namely 55 % for dry-snow conditions and 62 % for the entire season (dry- and wet-snow conditions).

Even for days without precipitation, the changes in total SWE can be quite large as can be seen in Figure 5 for days with $\Delta SWE_{ref} = 0$. For dry-snow conditions, the majority of the changes in total SWE on these days were within [-10 mm, +10 mm] for the 24 h period and within [-20 mm, +20 mm] for the 72-hour period.

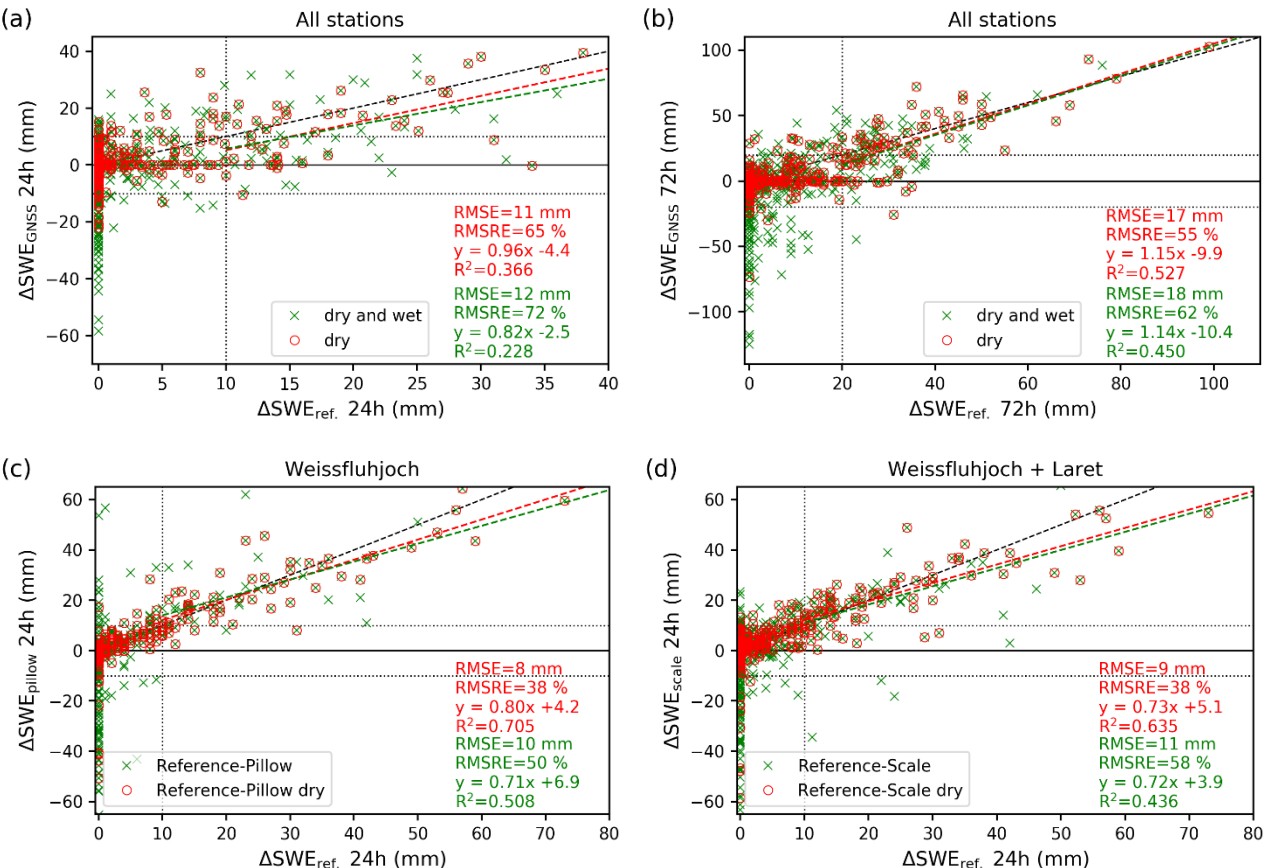

**Figure 5: Changes in GNSS-derived SWE vs. new snow water equivalent from reference measurements (pluviometer or observer) for all stations and for winter 2019-2020: (a) over 24 h and (b) over 72 h. Changes in SWE obtained with (c) snow pillow (Weissfluhjoch) and (d) snow scale (Weissfluhjoch and Laret) vs. reference measurements for both winters. The dashed lines indicate linear regression for the dry-snow conditions (red) and the entire season (green). The black dashed line indicates the 1:1 line. The linear regressions were computed only for data points with $\Delta SWE_{ref} > 10$ mm within 24 h or $\Delta SWE_{ref} > 20$ mm within 72 h (dotted lines).**

To evaluate the performance of the different methods with regard to new snow detection we compiled a contingency table
(Table 3) that compares the number of days with or without precipitation ($\Delta SWE_{ref,24h} \leq 10$ mm) with the number of days
with an increase, decrease or unchanged value of SWE from GNSS, scale and pillow. We used the same threshold ($\pm 10$ mm)
over 24 h for determining whether on a given day there was an increase ($\Delta SWE > 10$ mm), a decrease ($\Delta SWE < -10$ mm) or
no change ($|\Delta SWE| \leq 10$ mm). For the 72-h period we used a threshold of $\pm 20$ mm. On 28 out of 317 days (9 %) without
precipitation ($\Delta SWE_{ref} \leq 10$ mm) and dry-snow conditions, GNSS-derived $\Delta SWE_{24h}$ resulted in a false alarm (increase or
decrease). The magnitude of false alarms was up to $\Delta SWE_{GNSS,24h} = 32$ mm. On days with precipitation, 41 % (18 out of 44)
of the GNSS-derived changes were classified as no change or even decrease ($\Delta SWE_{GNSS,24h} \leq 10$ mm), i.e. these snowfall
events were missed. Missed events included snowfalls with up to $\Delta SWE_{ref,24\,h} = 34$ mm. For the 3-day sum of new snow,
there were fewer false alarms on days without precipitation, but again about 30 % of the precipitation days with
$\Delta SWE_{ref,72h} > 20$ mm were not detected. The maximum magnitude of the undetected events over 72 h was
$\Delta SWE_{ref,72\,h} = 32$ mm water equivalent. The maximum value for false alarms over 72 h was $\Delta SWE_{GNSS,72h} = 32$ mm.

If days with wet-snow conditions were included in the analysis the uncertainty of $\Delta SWE$ decreased compared to dry-snow
conditions, with an increase in false and missed precipitation days. Figure 5 shows that the increase in false events is strongly
influenced by melting (large increase of days with negative values). Therefore, the scatter of $\Delta SWE_{24h}$ was larger when wet-
snow conditions were included (Figure 5a, RMSRE = 72 %, and lower correlation). We did not find any distinct difference in
the uncertainty of $\Delta SWE$ between the sites at the different elevations.

Compared to the GNSS-derived data, the number of missed events for dry-snow conditions was much lower for the snow
pillow (17 %) and the snow scale (16 %) (Table 3). For dry-snow conditions, also RMSE and RMSRE of $\Delta SWE$ from pillow
and scale (Figure 5c,d) were moderately smaller than those of the GNSS-based $\Delta SWE$. However, the large deviation in SWE
of pillow and scale occurring at the onset of the melt season (see Figure 3) caused some large errors in $\Delta SWE$ under wet-snow
conditions.

**Table 3: Contingency table illustrating the detection performance of new snow events for GNSS (all sites, winter 2019-2020), snow pillow (Weissfluhjoch, 2018-2019 and 2019-2020) and snow scale (Weissfluhjoch and Laret, 2018-2019 and 2019-2020). We considered new snow days with an increase in ΔSWE$_{ref}$ larger than 10 mm in the preceding 24 h or larger than 20 mm in the preceding 72 h. For ΔSWE from GNSS, pillow and scale we defined three classes: (1) days with an increase if ΔSWE > 10 mm over 24 h, (2) days with no change if |ΔSWE| ≤ 10 mm and (3) days with a decrease if ΔSWE < -10 mm. For ΔSWE over 72 h we used ± 20 mm as threshold. N is the total number of days considered.**

| | | | Reference measurements | | | | | | | |
|---|---|---|---|---|---|---|---|---|---|---|
| | | | Dry snow | | | | All | | | |
| | | | ΔSWE 24 h | | ΔSWE 72 h | | ΔSWE 24 h | | ΔSWE 72 h | |
| | | | >10 mm | ≤ 10 mm | > 20 mm | ≤ 20 mm | >10 mm | ≤ 10 mm | > 20 mm | ≤ 20 mm |
| Prediction | GNSS | Increase | 26 | 19 | 48 | 11 | 45 | 44 | 78 | 26 |
| | | No change | 17 | 289 | 20 | 261 | 36 | 435 | 43 | 393 |
| | | Decrease | 1 | 9 | 1 | 6 | 1 | 75 | 2 | 74 |
| | | N | 44 | 317 | 69 | 278 | 82 | 554 | 123 | 493 |
| | Pillow | Increase | 48 | 11 | 78 | 4 | 70 | 32 | 108 | 21 |
| | | No change | 10 | 245 | 9 | 205 | 11 | 358 | 17 | 289 |
| | | Decrease | 0 | 7 | 0 | 10 | 0 | 68 | 3 | 79 |
| | | N | 58 | 263 | 87 | 219 | 81 | 458 | 128 | 389 |
| | Scale | Increase | 67 | 15 | 109 | 19 | 98 | 40 | 152 | 44 |
| | | No change | 13 | 336 | 13 | 269 | 17 | 544 | 30 | 440 |
| | | Decrease | 0 | 8 | 0 | 10 | 3 | 75 | 5 | 80 |
| | | N | 80 | 359 | 122 | 298 | 118 | 659 | 187 | 564 |

## 4.3 Liquid water content

The temporal evolution of the GNSS-derived LWC and the corresponding reference data for Weissfluhjoch, Laret and Klosters for the winter season 2019-2020 are shown in Figure 6. The reference data were obtained from the manual snow pit observations (capacitive probe). In addition, the snow temperature as was measured in the snow pit is indicated with three classes: dry (< 0 °C), partially dry (< 0 °C) and isothermal. For Weissfluhjoch also the LWC obtained from the upGPR data is shown (Figure 7a). In general, the higher the elevation the later liquid water was present or LWC was lower at a specific time within the season. Generally, we observed a good qualitative correspondence between the value of the GNSS-derived LWC and the snow cover temperature. The transition from dry to partially isothermal snow cover based on the snow temperatures corresponded with the first increase of LWC. The LWC was below 2 % for partially isothermal conditions and increased once the snow cover reached isothermal conditions. The periods with LWC < 1 % corresponded to periods with daily melting and freezing of the snow surface. Depending on the time of the manual snow temperature measurements, the snow cover temperature conditions were classified as dry (< 0 °C) or as partially isothermal. GNSS-derived values of LWC and measurements with the capacitive probe agreed well, yet values obtained with the capacitive probe were generally lower during periods with partially isothermal snow cover. We assume that the lower values are due to a systematic error of the dielectric measurement method, which is known to be subject to relatively large uncertainties, in particular at low values of LWC (Techel and Pielmeier, 2011). In contrast, the GNSS-derived and upGPR-derived values of LWC agreed well, with regard to both absolute values and variation in time for the period after 7 April 2020 (Figure 6a). For the preceding period since mid-February

2020, repeated melting at the snow surface was evident in the upGPR data, but the LWC could not be derived due to the limited resolution of the radar. The performance of LWC derivation with the GNSS method was similar at all sites and did not depend on elevation. The high values of LWC observed at the Weissfluhjoch site during the melt season 2020 were probably due to a particular snowpack layering with many ice lenses, which may have hindered melt water percolation. The LWC results for spring 2019 were similar to the above presented ones.

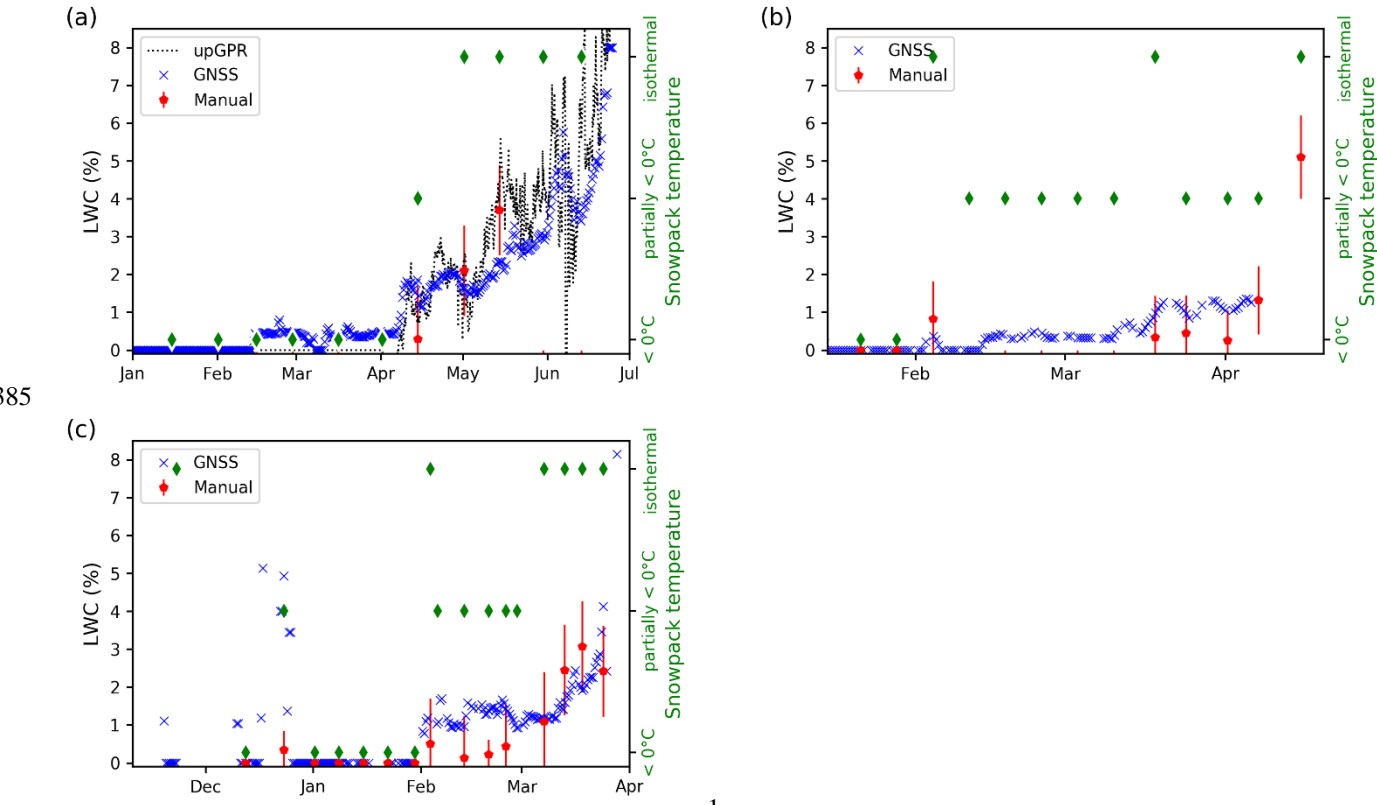

**Figure 6: GNSS-derived LWC during the winter 2019-2020 for (a) Weissfluhjoch, (b) Laret and (c) Klosters in blue. The red points show the manually measured LWC (capacitive probe) averaged over the depth and the vertical bars indicates the estimated error of the manually measured LWC. The green diamonds indicate the mean snowpack temperature. Only data points with HS > 5 cm**
**are shown.**

### 4.4   Snow depth

The seasonal evolution of the GNSS-derived HS and the reference data from the ultrasonic and laser sensors are shown in Figure 7 for the four sites and two winters. Both seasons followed the patterns as described for SWE in Section 4.1. Table 4 shows RMSE, RMSRE and linear regression values for the GNSS-derived HS relative to the reference values. Scatter plots of
GNSS-derived HS vs. reference data for all stations are shown in the Appendix D in Figure D1. Overall, GNSS-derived HS correlated well with the reference data; RMSE and RMSRE were 14 cm and 19 %, respectively, for all sites and both winters. The correlation for the high-alpine site Weissfluhjoch, where the dry-snow and wet-snow densification models were developed

and tested, was highest with R² = 0.99. RMSE values for all sites were in the range of 12 to 15 cm, without a clear dependence on elevation. However, the RMSRE increased with decreasing elevation – as was observed for SWE. Towards the end of the melt season 2018-2019, the GNSS-derived decrease in snow depth was delayed at Weissfluhjoch in June and at Laret and Klosters since mid-March. In contrast, the decrease was rather well captured during the melt season 2019-2020. At the lower elevation sites, the densification after a snowfall during dry-snow conditions was often overestimated. Moreover, small snowfalls on top of a thick snowpack were often not detected, in particular for wet-snow conditions (e.g. in February and March 2020 for Weissfluhjoch, Laret and Klosters).

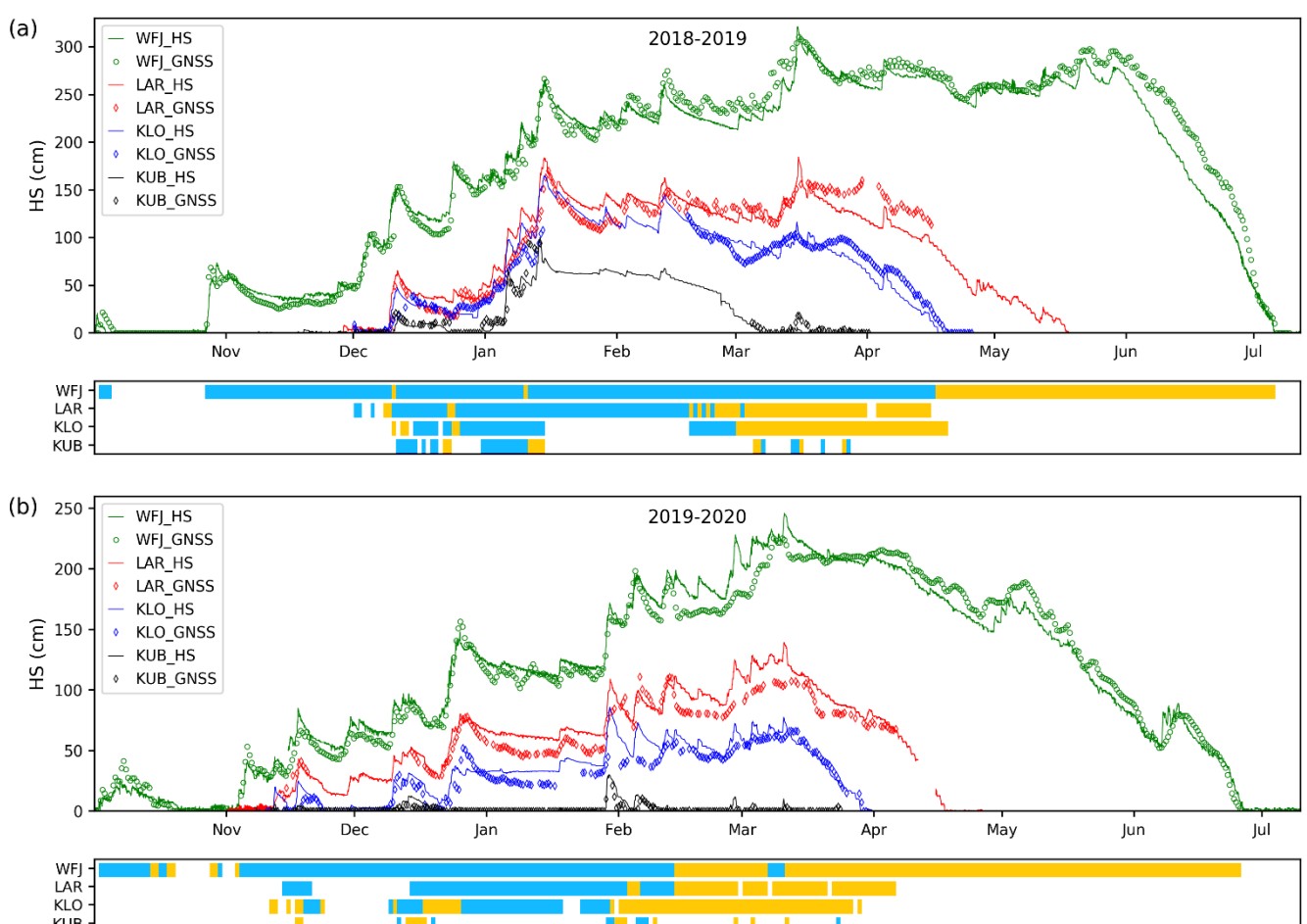

**Figure 7: GNSS-derived snow depth (HS) and reference data for (a) the winter 2018-2019 and (b) 2019-2020 for Weissfluhjoch 2540 m a.s.l. (WFJ), Laret 1510 m a.s.l. (LAR), Klosters 1185 m a.s.l. (KLO) and Küblis 820 m a.s.l. (KUB). The color bars indicate whether the dry-snow (blue) or wet-snow (orange) GNSS algorithm was used.**

**Table 4: Root mean square error (RMSE), root mean square relative error (RMSRE), number of data points N and $R^2$ from linear regression for GNSS-derived HS compared to the data from the automated sensors for all sites jointly and separately for the individual sites. Data points with HS < 10 cm were excluded from the analysis.**

| | | RMSE (mm) | RMSRE (%) | N (-) | $R^2$ (-) |
|---|---|---|---|---|---|
| **All** | | 14 | 19 | 1729 | 0.97 |
| | dry | 12 | 18 | 947 | 0.98 |
| | wet | 17 | 23 | 782 | 0.95 |
| **WFJ** | | 15 | 15 | 989 | 0.97 |
| | dry | 10 | 15 | 568 | 0.99 |
| | wet | 19 | 14 | 421 | 0.94 |
| **Laret** | | 15 | 18 | 383 | 0.88 |
| | dry | 14 | 18 | 227 | 0.94 |
| | wet | 16 | 18 | 156 | 0.78 |
| **Klosters** | | 12 | 26 | 337 | 0.83 |
| | dry | 13 | 25 | 138 | 0.90 |
| | wet | 12 | 26 | 199 | 0.76 |
| **Küblis** | | 12 | 45 | 48 | 0.79 |
| | dry | 17 | 41 | 36 | 0.86 |
| | wet | 21 | 57 | 12 | 0.83 |

## 5 Discussion

### 5.1 GNSS-derived snow cover properties and reference data

The GNSS-derived SWE values were accurate compared to the reference data and no particular dependence on the elevation of the sites or their differing local snow conditions were found, which implies that the algorithm is in addition to high-alpine sites also suitable for lower laying sites where snow conditions may be distinctly different. At lower elevations, there were more frequent changes between dry- and wet-snow conditions and more rain-on-snow events. Moreover, at lower elevations with a shallower snowpack the proportion of snow that is subject to daily melt-freeze cycles is larger than at sites with a thick snowpack, implying a larger impact on bulk snow cover properties.

Regarding all sites and the two winter seasons overall, the RMSE was 34 mm and the RMSRE 11 % compared to manual reference measurements, which also have an uncertainty of at least 5 % (López-Moreno et al., 2020; Royer et al., 2021). Previously reported findings on GNSS-based SWE measurements (Henkel et al., 2018; Koch et al., 2019; Steiner et al., 2019a) at Weissfluhjoch are in agreement with our results. For the three preceding winter seasons 2015-2016, 2016-2017 and 2017-2018, Koch et al. (2019) reported RMSE values of 41 mm for dry-snow and 73 mm for wet-snow conditions. These values are slightly higher than the ones we reported (29 mm and 62 mm, respectively) since Koch et al. (2019) used the SWE data from the snow pillow and scale as reference, which included an offset at the beginning of the melt season. Steiner et al. (2019a), using an alternative algorithm, reported an RMSE of 42 mm for dry-snow and 137 mm for wet-snow conditions at the Weissfluhjoch site for the season 2017-2018. In general, SWE and its temporal evolution over the entire winter season can

be captured very well with the GNSS method, which is very promising for, e.g. long-term monitoring of the snow cover and many hydrological applications.

The overall uncertainty of the other two automated SWE sensors, snow pillow and scale, was higher than the uncertainty of the GNSS-derived SWE. For dry-snow conditions, the RMSRE of the GNSS-derived SWE (11 %) was equal to the RMSRE of the snow pillow and slightly lower than REMSE of the snow scale (14 %). For wet-snow conditions, the uncertainty of the GNSS method (RMSRE = 11 %) was better than the pillow with an RMSRE of 15 % and the scale with an RMSRE of 20 %. The reported accuracies are in accordance with results from previous studies, which reported an uncertainty for the snow pillow

of 5-15 % (Serreze et al., 1999) and 8-21 % (Johnson et al., 2015). The higher uncertainty of the snow pillow and scale for wet-snow conditions is due to large deviations often observed at the beginning of the melt period. These anomalies are probably due to bridging effects caused by different heat fluxes at the bottom of the snowpack causing different melt rates and snow densification above the sensor surface compared to the surrounding ground, and by meltwater infiltration and drainage (Johnson and Schaefer, 2002; Johnson et al., 2015).

The results for LWC derived from the GNSS data were in accordance with the reference data and within the uncertainty of the reference data (0.5-1%; Fierz and Föhn, 1995; Mavrovic et al., 2020). This finding suggests that LWC can be measured reliably also for snow conditions different from those found at Weissfluhjoch where the method was developed (Koch et al. 2014). The quality of LWC data derived from GNSS was found to be similar to those derived with the upGPR method according to Schmid et al. (2014). However, the GNSS method does not need independently measured data from another source and supervision in

the data processing such as snow surface picking in radargrams (Schmid et al., 2014). Therefore, the GNSS method is well suited for operational monitoring of LWC. As it can measure LWC non-invasively from below the snow cover, it could be used for wet-snow avalanche research and forecasting. The frequency of data sampling of 12 h used in this study did not allow to reveal the sub-daily wetting and refreezing cycle. However, LWC derivation at (half-)hourly frequency is possible and allows detecting sub-daily melt-freeze cycles as demonstrated by Koch et al. (2014) and Schmid et al. (2015).

The GNSS-derived snow depth data, which can be seen as a by-product of the SWE derivation, showed a good correlation with the reference data and an acceptable uncertainty. For the lower elevation sites, the densification after a snowfall event was often too fast because the exponential densification rate for dry-snow conditions we used does not apply equally well to all situations. Moreover, small snowfalls on top of a thick snowpack in spring were often not detected indicating that the density model for wet-snow needs to be improved for such conditions. The quality of the GNSS-derived HS is, however, not

comparable to the well-established and widely used ultrasonic or laser HS sensors as the implemented simple snow density models cannot capture the HS evolution for each snowfall event and densification situation reliably. Therefore, the GNSS-derived HS is currently only of interest for operational application in the case of a stand-alone installation of a SnowSense® GNSS station. It is a supporting value for the other snow cover parameters SWE and LWC during wet-snow conditions, whereas for dry-snow conditions it is just a model output relying on the GNSS-derived SWE. Therefore, future efforts should

aim at improving the densification model used for HS derivation with the objective of further improving the accuracy of SWE and LWC. A qualitative evaluation of the rain-on-snow events showed no major influence on both SWE and HS and a moderate

increase in LWC for some events (Appendix B). The small number of available datapoints does not allow conclusions on the influence of rain-on-snow events on the GNSS-derived SWE or HS. More research in this regard would be needed; however, we can at least exclude that large effects occur for events of the observed magnitude (< 30 mm in 24 h) as those should
otherwise be visible in the figures in Appendix B.

## 5.2    Current limitations in retrieving the water equivalent of new snow

While monitoring the seasonal evolution of snow cover properties is valuable for various climatological and snow hydrological applications, other applications require an exact estimation of variations in SWE at a shorter time scale. Currently, the GNSS-derived SWE shows significant daily fluctuations resulting in a rather low accuracy in the estimation of precipitation
accumulated over 24 and 72 h. The GNSS-derived changes in SWE were in general related to the reference precipitation data, but scattered largely compared to the reference data. Measuring small daily variations on top of the much larger total SWE is quite challenging and emerges in the observed large relative errors (RMSE) for $\Delta$SWE. During the melt season, many of the negative deviations can be explained by the fact that snow melting is not reflected in the reference data. False positive events (increase in SWE) and decreases in SWE during the dry-snow conditions were mainly due to uncertainties in the GNSS-
derived SWE determination caused by increased measurement noise and multipath propagation leading to an erroneous integer ambiguity fixing. In addition, snow drift by wind and the resulting spatial variability may be a source of uncertainty in the daily variations of SWE for both, the GNSS-derived and the reference data. Moreover, pluviometers are known to be prone to under catch of up to 50 % due to wind during snowfall (Grossi et al., 2017; WMO, 2019). Therefore, errors in the reference data may as well contribute to the observed large deviations. However, wind speed was generally low at the lower elevation
sites and, therefore, little influence on SWE and precipitation measurements is expected. For the Weissfluhjoch site, we used manual data as reference since these are less influenced by wind. Moreover, cumulated precipitation data agreed well with SWE for all sites.

A correct carrier phase integer ambiguity resolution is necessary for accurate SWE determination since an error of only one cycle leads to a considerable bias in the SWE estimate. The integer ambiguity fixing of GNSS measurements below snow is
challenging since the pseudo-range measurements are affected by severe multipath propagation and since both the integer ambiguities and the snow-caused time delay are nearly constant over short time periods, i.e. the parameters can only be separated based on the change of the satellite geometry over time. As the orbital period for GNSS satellites is nearly 12 hours and as the integer ambiguity fixing uses the SWE estimates from the previous day as prior information, an erroneous integer fixing may occur over subsequent days.
Missed snowfall events and false alarms as described regarding GNSS-derived changes in SWE are, as the SWE and HS derivation are interconnected, also visible in the time evolution of the GNSS-derived HS (Figure 7). In particular in spring, rather small snowfall events on top of a thick snowpack are mostly not detected at first, although in the following days HS increases progressively.

In summary, we conclude that for practical applications such as avalanche forecasting the GNSS-derived daily changes in
SWE are not sufficiently reliable and accurate. The currently necessary data measurement period for the SWE derivation of at
least 6 h is a further limitation for such (sub-)daily applications, which need hourly input data. On the other hand, snow pillow
and scale allow a real time observation of precipitation events. In fact, for dry-snow conditions, the performance of the snow
pillow in determining changes in SWE over 24 h and 72 h was better than with GNSS. However, for wet-snow conditions,
both, the scale and the pillow were unreliable due to the large errors caused by bridging effects and other artefacts.

## 5.3  Stability of GNSS-derived snow parameters regarding data gaps


A measuring system meant for operational use does not solely need to deliver accurate data but also to be reliable in terms of
operation. This is particularly challenging for sensors systems that are subject to harsh conditions and often not accessible for
maintenance due to remoteness or dangerous access, e.g. in case of avalanche danger. As described in Section 3, our GNSS
data series over two winter seasons had some data gaps. The unusual large snow load in January 2019 caused the failure of the
mast at the sites in Klosters and Küblis and consequently data loss. This clearly shows that it is crucial that the reference
antenna is always mounted on a stable existing structure or massive pole well anchored to the ground. Further data gaps were
caused by problems with the initial version of the power management firmware. These problems could be fixed by a firmware
update in summer 2019.

However, data gaps can occur also with the best measurement design, for instance, due to a power shortage an operational
measurement system should resume uninfluenced by the interruption. It is therefore crucial that data quality is not affected by
data gaps. The algorithm deriving the snow cover properties from the GNSS signals could particularly be prone to such
problems since it recursively derives all snow properties from the previous data for wet-snow conditions, whereas for dry-
snow conditions only HS depends on previous data. Therefore, we analyzed the consequences of data gaps. We chose
exemplarily the data gap from 16-23 January 2020 at the site Klosters, which occurred due to corrosion. We investigated the
impact on the derivation of the snow cover properties for the entire period after the data gap occurred, which was mainly
characterized by the wet-snow period until the end of April 2020. During these 7 days of lacking GNSS data, there was a
snowfall with 10 cm new snow ($\Delta$SWE = 8 mm) (Figure 8). The parameter derivation after this data gap was implemented in
a post-processing step in three different ways: (1) neglecting any previous information, which is normally stored after
processing, corresponding to a cold start of the system, (2) using the HS information of the last data point of snow cover
properties as input and (3) using the HS value measured with the laser sensor as a priori information.

The GNSS-derived SWE was affected only minimally no matter which of the three methods was used. For HS the differences
between the three approaches were large. For LWC also significant differences existed as it is calculated based on HS and
signal strength. The best solution was obtained with the a priori information from the HS laser sensor. In case we used the last
available data point, HS was underestimated by approximately the amount of snow fallen during the data gap. This offset
propagated more or less constantly for the rest of the season. If no previous information was used, HS was largely
overestimated since with a cold start the algorithm erroneously assumed an initial snow density of 100 kg m$^{-3}$ for the entire

snowpack. As the LWC calculation within the algorithm depends besides GNSS signal strength information also on the GNSS-derived HS, it therefore reflects to a certain extent also the error in HS. Thereby, the error in LWC increased with a decrease in HS and an increase in signal strength. This example shows that SWE, being the main target value of the GNSS approach,

was only affected marginally by an error in HS or LWC and would not be affected at all during dry-snow conditions, as the derivation solely depends on carrier phase measurements and no additional changes in the snow cover parameters. Therefore, we conclude that the SWE derivation is robust with regard to data gaps. Regarding HS and LWC, however, a more complex model of densification or an HS estimate obtained by GNSS reflectometry may mitigate the problem (see below in Section 6.1.) Moreover, a potential further improvement may be to feed the onboard processing in real time with updated a priori

information on e.g. SWE and snow depth after a data gap so that also HS and LWC can be derived reliably.

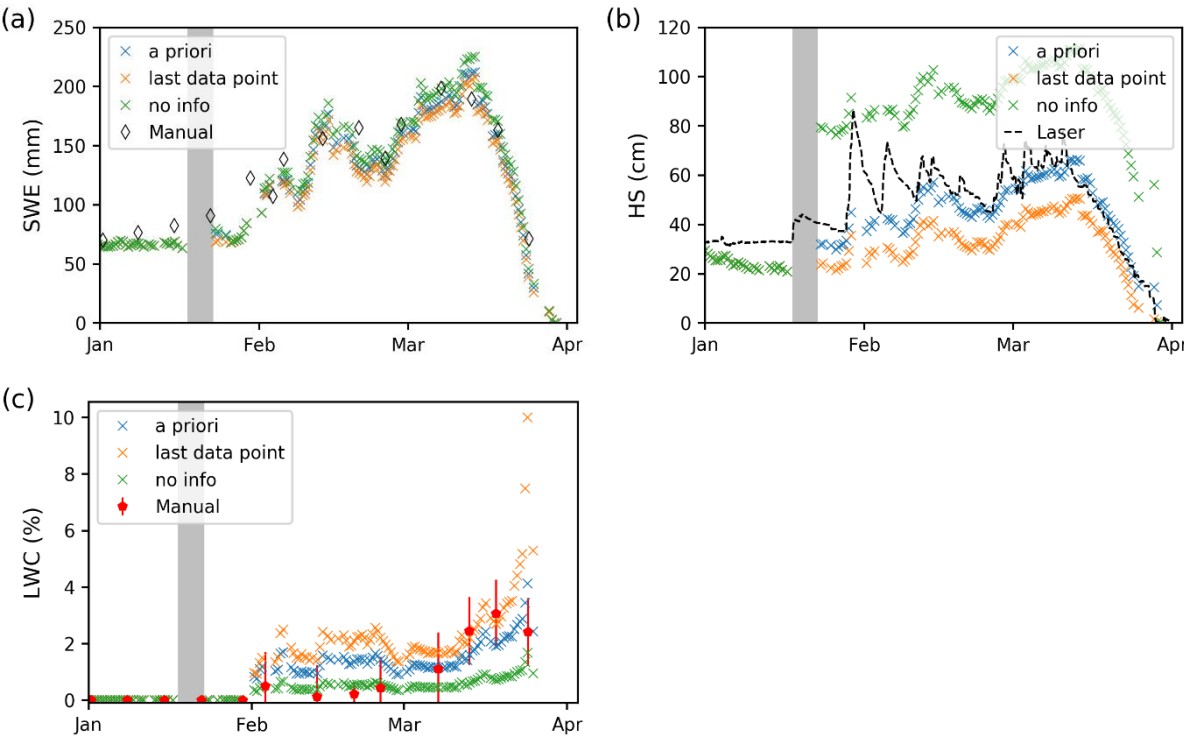

**Figure 8: GNSS-derived (a) SWE, (b) HS and (c) LWC after a data gap of 7 days with corresponding reference value at the site Klosters for the winter period 2020. Three approaches were used to derive the snow parameters after the data gap: neglecting any previous information (green), using the last available HS data point (orange) and using the reference HS measurement as a priori**

**information (blue).**

# 6 Outlook on potential improvements and further applications

## 6.1 Density and snow depth estimation

Although improvements of the dry- and wet-snow density models and the HS estimation is out of the scope of this paper, we would like to outline potential future developments. As the densification is rather too fast at the lower elevation sites, changes of the set time period of densification, which was in our case 30 days, the applied exponential densification rate as well as the set maximum dry snow density could be optimized for the layer-dependent exponential dry-snow model. For instance, we assume that the densification rate decreases the shallower the snowpack, as the bulk weight of the snowpack decreases. The simple wet-snow densification approach, which is up to now solely dependent on LWC and SWE changes, could be improved by including additionally an exponential layer-dependent snow densification similar to the dry-snow model. Such optimizations seem feasible based on the data collected in this study.

Alternatively, models deriving SWE statistically from HS and considering elevation, region and season could be integrated (e.g., Jonas et al., 2009; Winkler et al., 2021). Such models show good results as they rely on numerous HS measurements at various locations and over long time periods. However, regarding an implementation into the GNSS algorithm, they would first need to be inverted so that HS could be derived from SWE and an ad hoc calibration would be necessary for each climatic region.

However, some effects such as a significant decrease in snow densification over time due to temperature gradient driven snow metamorphism leading to the development of faceted crystal and depth hoar (e.g., Wiese and Schneebeli, 2017), typically occurring for shallow snowpacks in cold conditions, could still be difficult to capture with the above mentioned methods. Such a situation with a shallow snowpack and low temperatures leading to faceting and slowing down snow settlement was observed in Klosters in January 2020. Therefore, we suggest that a combination of the applied GNSS approach with GNSS reflectometry (e.g. Larson et al., 2009) may lead to a more stable HS derivation, as it would allow tracing the densification rate after a snowfall event. Reflectometry approaches derive HS via exploiting the multipath of reflected signals at a GNSS antenna above the snow cover, which could be in our case the reference antenna. Recent studies showed a high accuracy of HS derivation applying GNSS reflectometry (Boniface et al., 2015; Zhang et al., 2020) and the possible use of low cost GNSS receivers (Rover and Vitti, 2019). With such a combined GNSS signal delay, attenuation and reflectometry approach, all snow cover properties could solely be derived from GNSS signals.

## 6.2 Potential further applications and improvements of the GNSS algorithm

The GNSS-based snow parameter determination is suitable for many applications including hydrology, snow load monitoring and avalanche forecasting. In addition, the GNSS-based snow cover properties could be used for validation of new promising active microwave remote sensing approaches deriving snow height and SWE under dry- and wet-snow conditions at scales of 100 to 250 m in mountainous regions (Lievens et al., 2019; Lievens et al., 2021; Tsang et al., 2021).

With the GNSS method snow cover properties are measured non-invasively from below the snow cover with a small GNSS antenna. Therefore, the ground antenna could be installed in avalanche terrain without the risk of being damaged by avalanches provided the reference antenna is mounted at a safe location, e.g., on a nearby ridge. In general, the antenna below the snow and the reference antenna can be separated by several kilometers in horizontal direction and by up to 100 m in vertical direction without the need to consider differential atmospheric errors provided the overall meteorological conditions do not differ. Measuring LWC is relevant for studying wet-snow and glide-snow avalanches. However, some adaptations of the GNSS algorithm and data validation are needed, e.g. for on-slope measurements, since the present GNSS system was developed for flat terrain with the purpose of retrieving SWE in remote areas for hydrological applications.

The focus of our future work will be on the reduction of fluctuations to improve the determination of SWE and water equivalent of new snow, as discussed in Section 5.2, as well as the reduction of the measurement period for GNSS-derived snow parameters. We see mainly three opportunities: 1) The use of all 4 GNSS (GPS, Galileo, Glonass and Beidou) compared to the current GPS/Galileo dual constellation solution. The integration of Beidou is straightforward but the integration of Glonass needs to consider a Frequency Division Multiple Access (FDMA) adapted ambiguity resolution technique. 2) The integration of additional frequencies (L2, L5/ E5, E6) compared to the current single-constellation solution. The first dual-constellation mass-market GNSS receivers have recently become available and it is expected that mass-market GNSS receivers will be able to track all frequencies including E6 in the near future. 3) The use of a Kalman filter with an integrated integer least-squares estimator instead of a least-squares estimation. Initial results show that, with these three improvements, the measurement period can be significantly reduced to less than one hour. Hourly input data would be particularly beneficial for an accurate determination of the water equivalent of new snow and in general of sub-daily changes of SWE that are crucial for avalanche as well as flood forecasting.

## 7    Conclusions

We installed GNSS snow measurement systems at four sites along a steep elevation gradient (820, 1185, 1510 and 2540 m a.s.l.) in the eastern Swiss Alps for two winter seasons (2018-2020) and compared the GNSS-derived snow cover properties with concurrent reference data.

The GNSS-based SWE measurement was robust and accurate. We did not observe any notable dependency on elevation or snow conditions. The uncertainty was similar for dry-snow and wet-snow conditions and was negligibly influenced by rain-on-snow events. Compared to manual reference measurements, considering the data from all sites jointly, the RMSE was 34 mm and the RMSRE was 11 %. This uncertainty was achieved for a GNSS data frequency of 12 h. The shallower the snowpack was, the larger became the relative error. Therefore, SWE values below 10 mm could not accurately be determined. Still, the GNSS method reliably detected whether snow was lying on the ground or not. The uncertainty of GNSS-derived SWE was similar to the uncertainty of SWE measurements obtained with snow scale and pillow for dry-snow conditions and higher for wet-snow conditions. However, noise in the GNSS-derived SWE prevented a reliable estimation of the mass of newly fallen

snow during 24 h and 72 h. Only large snowfall events were detected, but still with poor accuracy in SWE changes. Snow scale and pillow showed better results in this regard under dry-snow conditions but performed poorly under wet-snow conditions. Currently, these methods are not suitable for reliably and accurately estimating the water equivalent of new snow for practical applications such as avalanche forecasting. Regarding the GNSS algorithm, further developments may overcome this deficiency.

The derivation of LWC was robust and the values of LWC were in the range of the manual and upGPR measurements. The GNSS method seems suitable for continuous LWC determination, which could be of interest for wet-snow avalanche forecasting.

As a by-product, the GNSS-derived HS showed in general a good correlation to the reference values with a RMSE of 14 cm and RMSRE of 19 %. However, snow densification after a major snowfall especially during dry-snow conditions was generally

too fast at the lower elevation sites. Moreover, with a thick snowpack during wet-snow conditions, snowfall events were not captured with the currently implemented simple wet-snow densification model. Future improvements of the dry-snow and wet-snow densification model might mitigate these problems.

Overall, our analysis showed that the GNSS system can reliably measure the seasonal evolution of SWE at different elevations where different snow conditions prevail. Hence, the GNSS-based derivation of SWE is suited for operational SWE monitoring

and a valuable alternative to manual measurements or other automated SWE sensors. Moreover, the GNSS method represents, to the best of our knowledge, the most appropriate and cost-effective approach for measuring SWE and LWC simultaneously, continuously and non-destructively.

**Appendix A: Seasonal snow density evolution**

In Figure A1 we show the seasonal evolution of the snow density for all stations in relation to SWE. The density was initially low and increased in general with increasing SWE, although some larger snowfall events caused the density to temporarily decrease. Toward spring, SWE decreased with the density staying high. The maximum density was higher at higher elevations due to the larger amount of snow accumulated over the season. In the second season (2019-2020) there was less snow and generally lower values of snow density and SWE were observed. Figure A2 shows the mean, maximal and minimal air

temperatures at the four sites. The air temperature was generally higher at lower elevations (excluding some inversion effects e.g. at Klosters and Laret in January 2020).

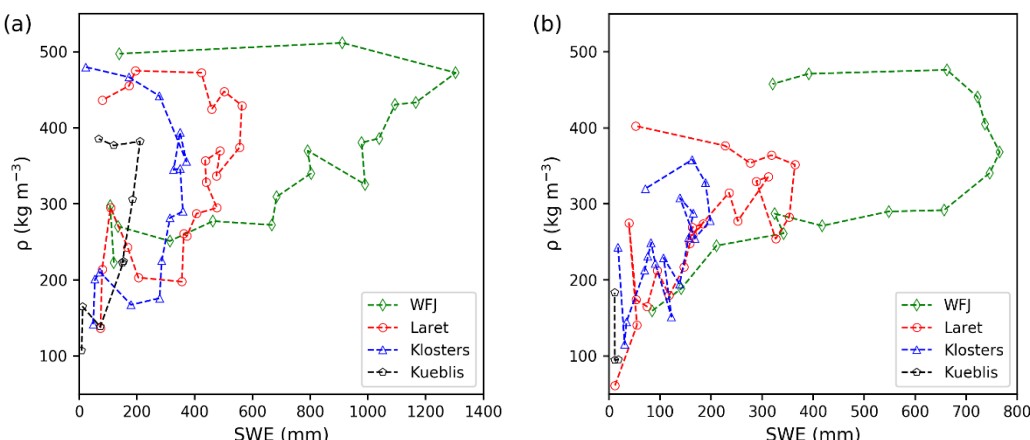

**Figure A1: Seasonal evolution of snow density vs. SWE from manual measurements for (a) 2018-2019 and (b) 2019-2020 at the four sites Weissfluhjoch 2540 m a.s.l., Laret 1510 m a.s.l., Klosters 1185 m a.s.l. and Küblis 820 m a.s.l. Measurements were done weekly**
**at the three lower elevations sites and bi-weekly at Weissfluhjoch.**

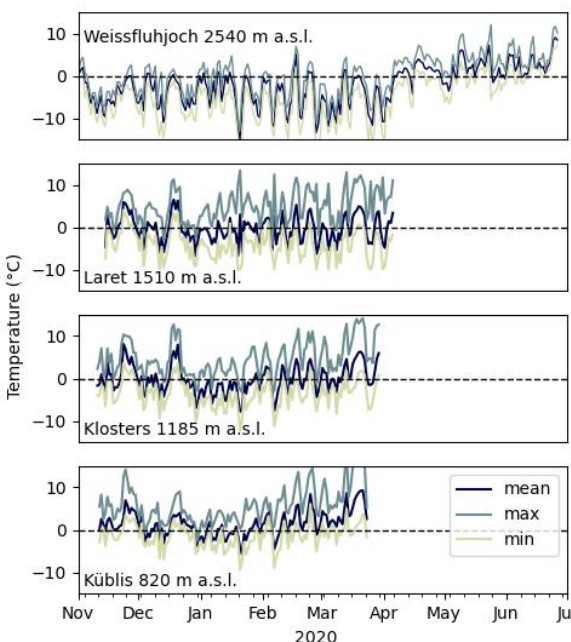

**Figure A2: Mean, maximal and minimal daily air temperature for 2019-2020 at the four sites Weissfluhjoch 2540 m a.s.l., Laret 1510 m a.s.l., Klosters 1185 m a.s.l. and Küblis 820 m a.s.l. The maximum temperatures at the Klosters and Küblis may be generally too high since we used unventilated sensors.**

### Appendix B: Rain-on-snow events

Figure B1-4 show the frequency and magnitude of the rain on snow events at the four measurement sites. The site in Klosters showed the largest frequency of rain-on-snow events throughout the winter season, whereas at higher elevations the rain-on-snow events were concentrated early in the season and in spring well into the melting phase when the snowpack was already wet. At the Küblis site a large amount of winter precipitation fell in form of rain or as a combination of snow and rain over 24 h. The robustness of the GNSS-derived snow parameters during rain-on-snow events is demonstrated with Figure B1. We did not observe any considerable effect of rain-on-snow events on the GNSS-derived SWE or HS. The LWC increased during some of the larger rain-on-snow events. Moreover, the cumulated precipitation (pluviometer) agreed well with the values of SWE weekly measured for the dry-snow part of the season provided melting early in the season is neglected, as occurred in 2019-2020 at the Klosters site.

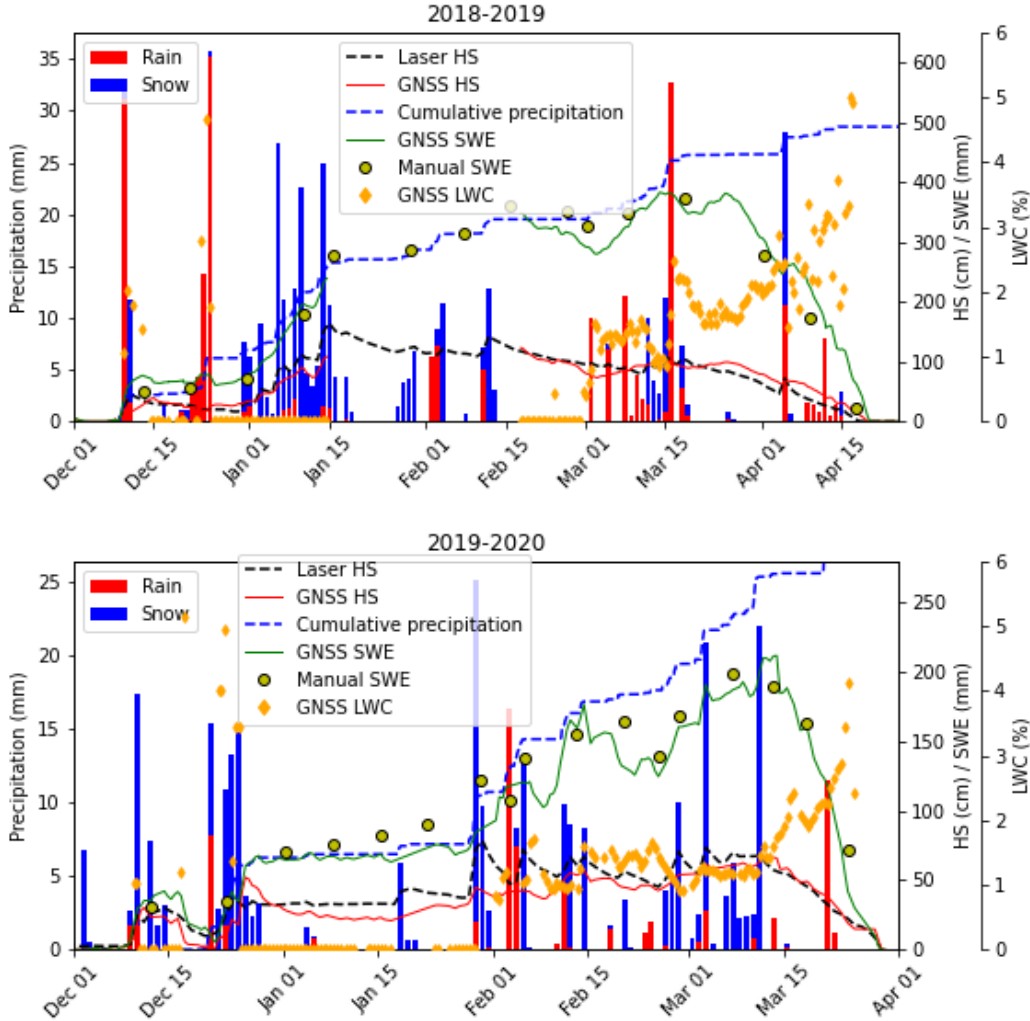

**Figure B1: Seasonal evolution of GNSS-derived SWE and HS and corresponding reference data for Klosters. The blue columns correspond to precipitation in form of snow; the red columns correspond to rain. Reference precipitation was measured by a pluviometer and classified as rain for air temperature T > 1.1 °C and as snow for T ≤ 1.1 °C. The various rain-on-snow events did not affect GNSS-derived SWE and HS.**

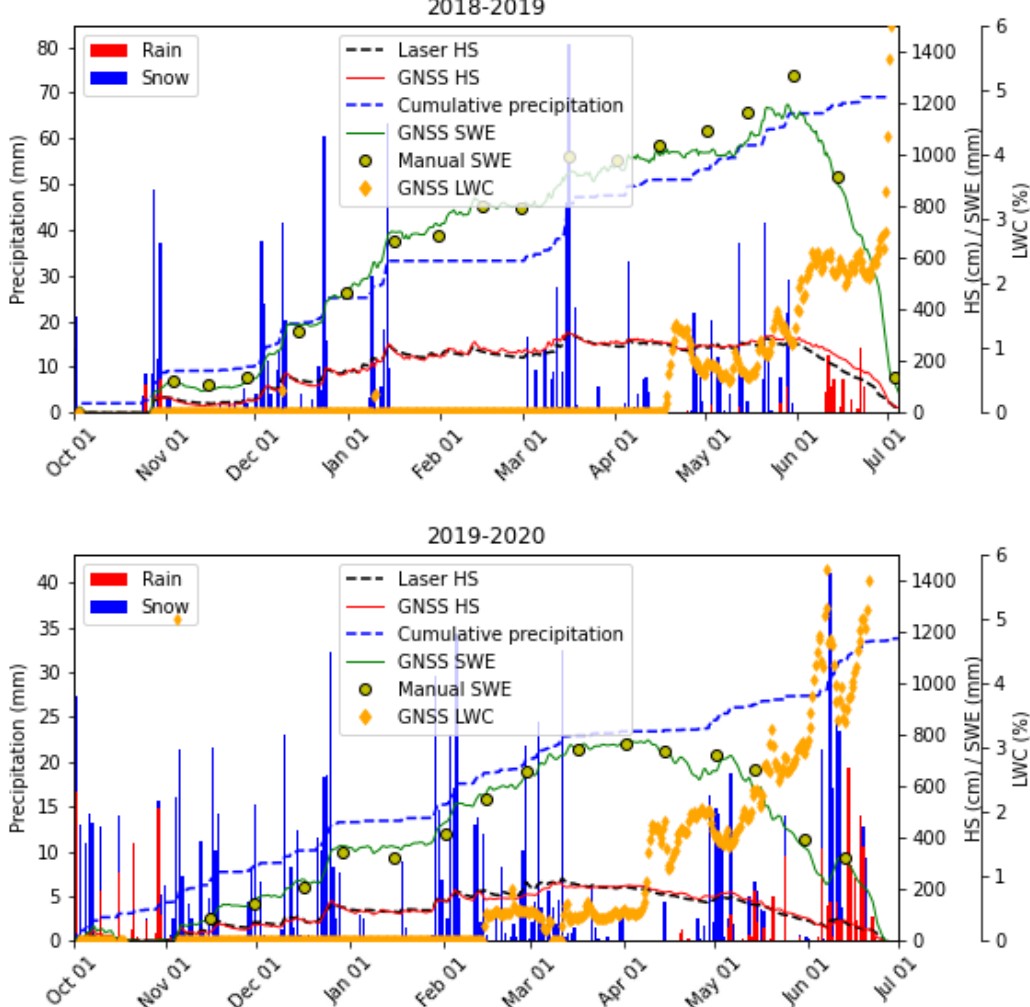


**Figure B2: Seasonal evolution of GNSS-derived SWE and HS and corresponding reference data for Weissfluhjoch. The blue columns correspond to precipitation in form of snow; the red columns correspond to rain. Reference precipitation was measured by a pluviometer and classified as rain for air temperature T > 1.1 °C and as snow for T ≤ 1.1 °C.**

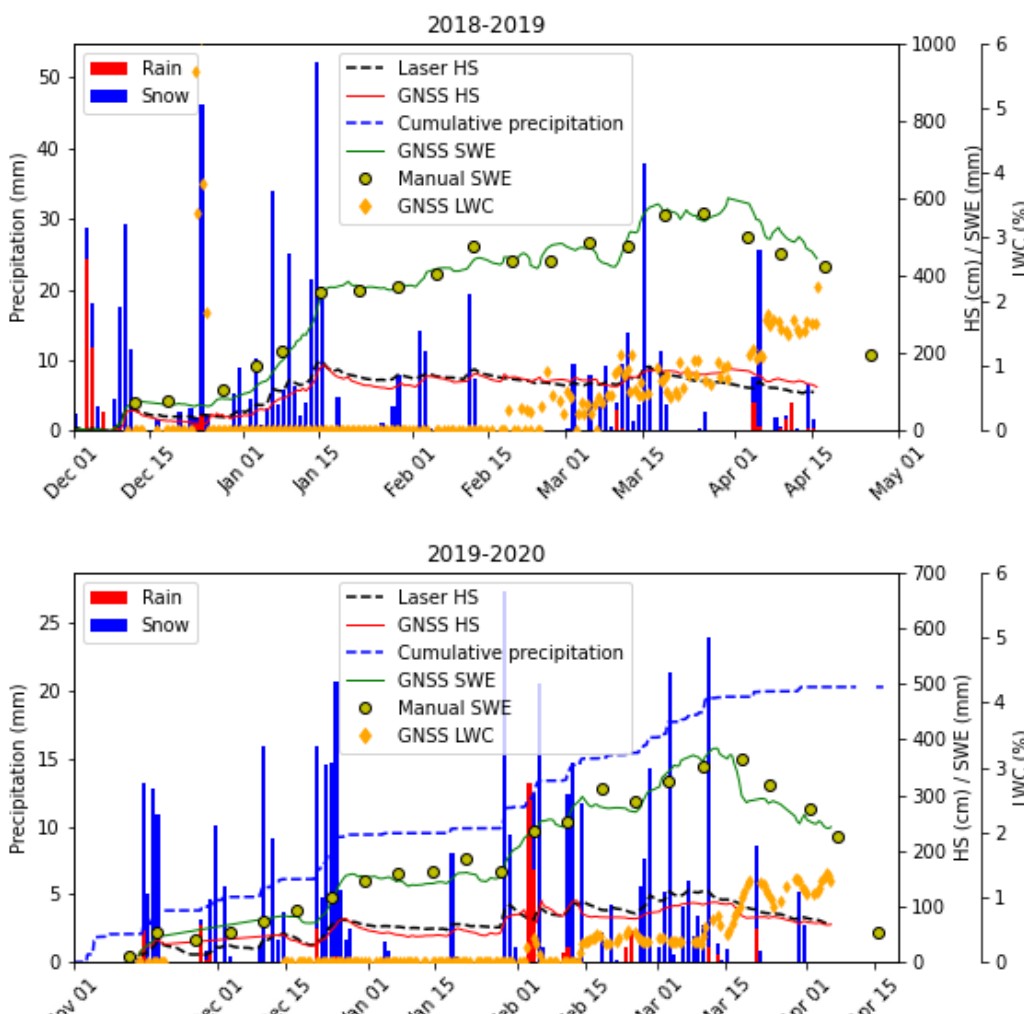


**Figure B3: Seasonal evolution of GNSS-derived SWE and HS and corresponding reference data for Laret. The blue columns correspond to precipitation in form of snow; the red columns correspond to rain. Reference precipitation was measured by a pluviometer and classified as rain for air temperature T > 1.1 °C and as snow for T ≤ 1.1 °C**


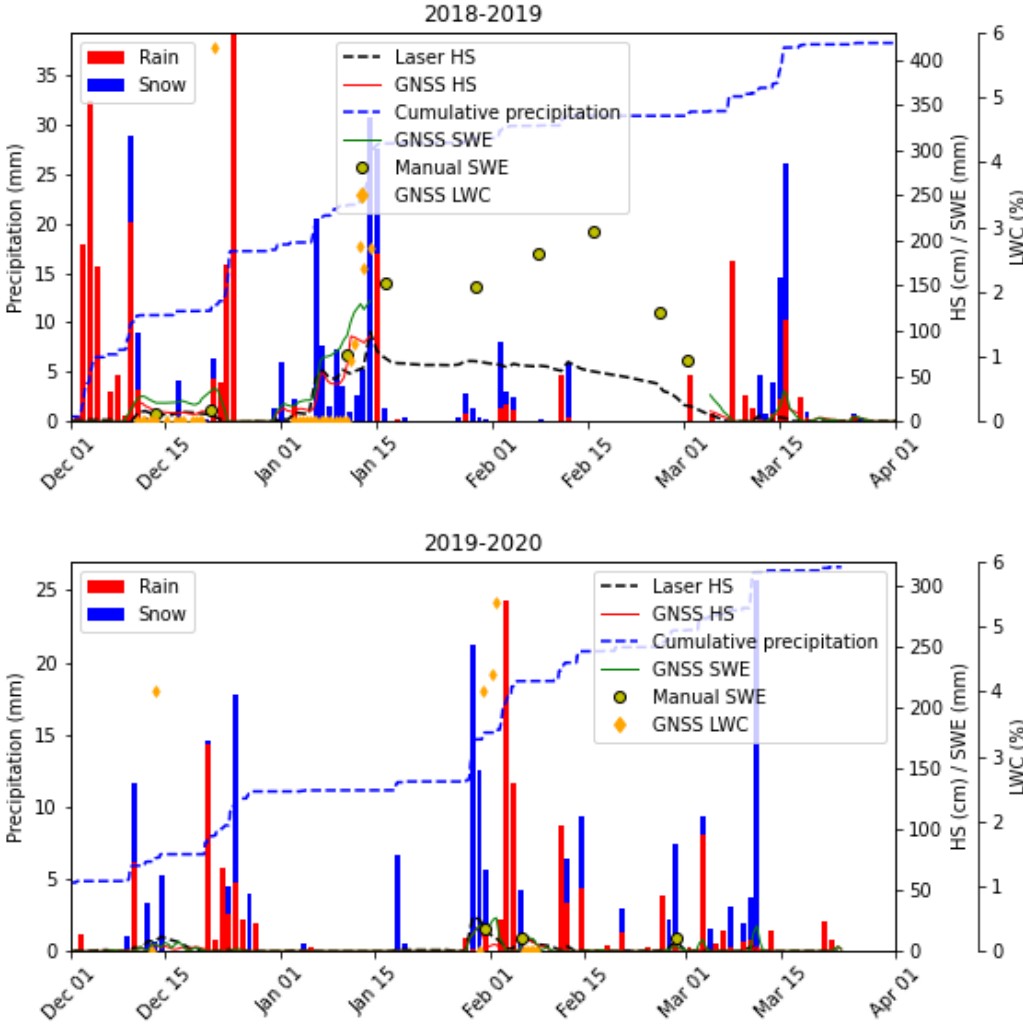

**Figure B4: Seasonal evolution of GNSS-derived SWE and HS and corresponding reference data for Küblis. The blue columns correspond to precipitation in form of snow; the red columns correspond to rain. Reference precipitation was measured by a pluviometer and classified as rain for air temperature T > 1.1 °C and as snow for T ≤ 1.1 °C.**

**Appendix C: Detection of snow on the ground**

Figure C1 illustrates the ability of the GNSS signal-based method to discern whether snow is lying on the ground for HS > 5 cm and SWE > 5 mm. However, the absolute values of GNSS-derived HS differ largely from HS measured with the laser sensor. It is to be mentioned that for such low amounts of snow the spatial variability in HS may be high, limiting the validity of the

comparison.

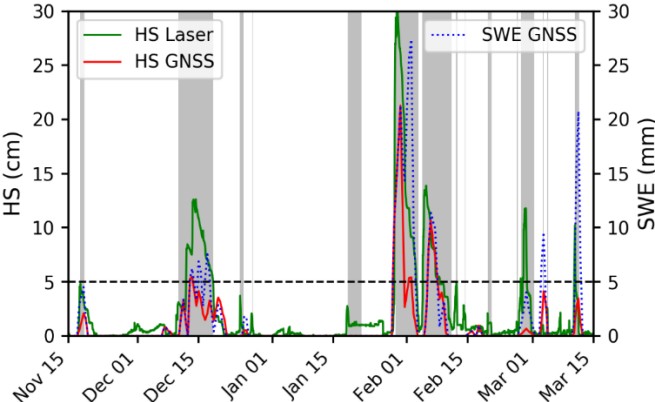

**Figure C1: GNSS-derived SWE and HS from laser sensor measurements at Küblis for the winter season 2019-2020. The gray zone indicates when snow was covering the GNSS ground antenna as determined from concurrent webcam pictures. It can be seen that for HS > 5 cm and SWE > 5 mm (horizontal dashed line) the GNSS system could discern well if snow was laying on the ground.**


## Appendix D: Snow depth validation

In addition to Figure 7 and Table 4, we show scatter plots of GNSS-derived HS vs. reference data for all stations in Figure D1.

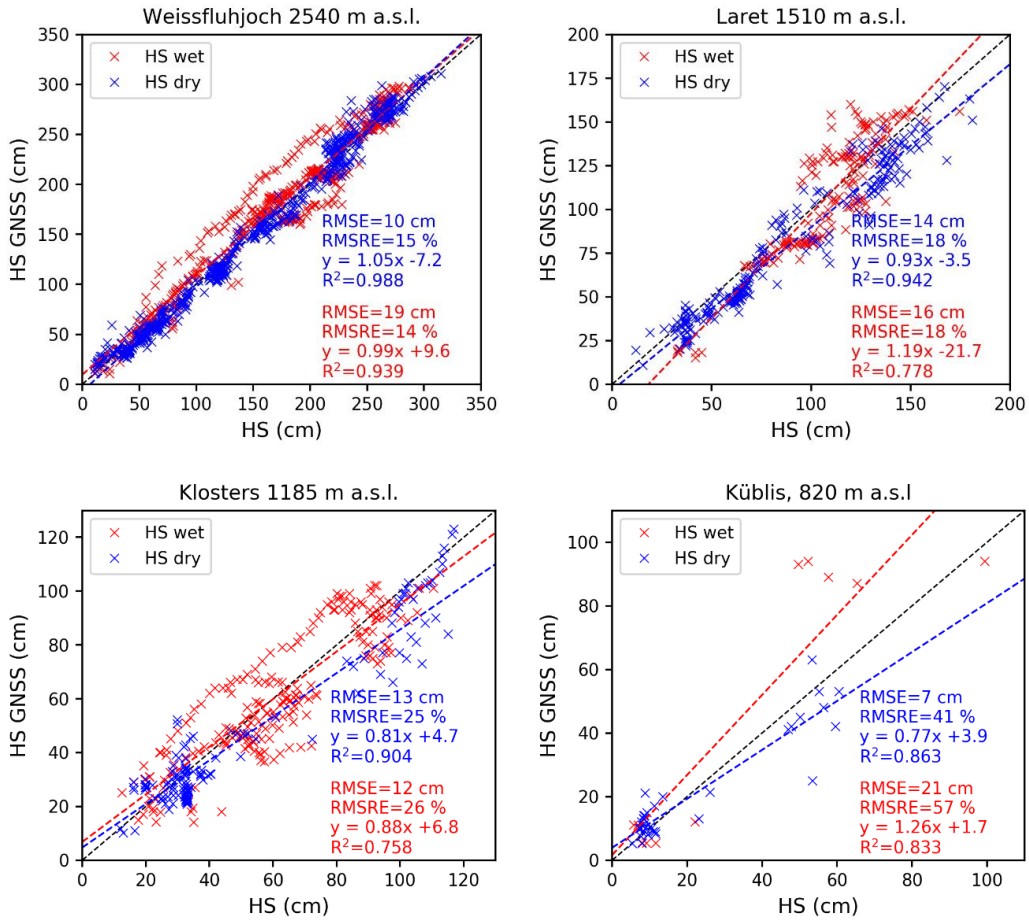

**Figure D1: Scatter plot of GNSS-derived snow depth (HS) vs. automatic measurement with ultrasonic and laser sensors for dry-snow conditions (blue) and wet-snow conditions (red). The dashed lines represent the linear regressions. Data points with HS < 10 cm were excluded from the analysis.**

## Data availability

The essential data are available on the WSL data portal Envidat at https://doi.org/10.16904/envidat.186 (Capelli et al., 2020).

## Author contribution

The study was designed by JS, CM, AC, FK and PH. ML, PH, AC and FK installed the SnowSense® GNSS stations in Laret, Klosters and Küblis; the instrumentation for the reference data was additionally installed in Klosters and Küblis. The GNSS

sensors at WFJ were already setup before the runtime of this study by FK, but were updated by ML, PH, AC and FK in 2018. AC and CM collected the reference data. Data curation of the GNSS data was done by FK. AC visualized and analyzed the data. AC and FK prepared the manuscript with contributions from all co-authors.

**Competing interests**

Achille Capelli, Franziska Koch, Christoph Marty declare they have no competing interests. Patrick Henkel, Markus Lamm and Florian Appel are related to the commercialization of the GNSS measurement setup as SnowSense® GNSS snow monitoring stations. Jürg Schweizer is a member of the editorial board of the journal.

**Acknowledgements**

This project was funded by MeteoSwiss in the Framework of GCOS Switzerland. The development of the SnowSense® GNSS
algorithms and station solution was a product of the ESA business applications demo project SnowSense (ESA Contract No. 000113149/14/NL/AD). Furthermore, we thank MeteoSwiss for the precipitation data, Peter Warnier for the snow measurements at the Klosters site, Charles Fierz for valuable discussions, and the members of the Snow Physics Group at SLF, in particular Henning Löwe and Matthias Jaggi, for the measurements data from the Laret site. We thank the reviewers A. Royer and A.N. Arslan for the constructive and valuable comments that helped to improve the manuscript.

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
