# Peer review of "GNSS signal-based snow water equivalent determination for different snowpack conditions along a steep elevation gradient"

_The Cryosphere, 2021_

## Referee Comment (RC1)

https://doi.org/10.5194/tc-2021-235
**GNSS signal-based snow water equivalent determination for different snowpack conditions along a steep elevation gradient**
 by Capelli et al.

This paper completes the evaluation of the GNSS-derived approach for SWE monitoring using the retrieval algorithm already presented by Koch et al. (2019) and validated at the high-alpine site Weissfluhjoch (2540 m a.s.l), with data at 4 altitudes in the Alpes (820, 1185, 1510 and 2540 m a.s.l.). The performance of this approach is thus assessed for shallow to deep snowpack, with more frequent changes between dry- and wet-snow conditions at low altitude, potential differences in densification and a higher influence of rain events compared to the high-alpine site Weissfluhjoch (2540 m a.s.l).

This article first presents the uncertainty results for the Snow Water Equivalent (SWE), the Liquide Water Content (LWC) and the snow depth (HS) estimates derived from SWE and LWC retrieved data for each of the 4 study sites.

The authors then analyzed the potential detection of snow variations over a short period of time (24 h and 72 h) by comparing these with reference precipitation data, and discussed the current limitations in retrieving new snow.

Since the retrieval of HS estimates and LWC parameter are derived using a recursive process from previously retrieved data, the authors assess also the stability of GNSS-derived snow parameters regarding data gaps.

Lastly, outlook on potential improvements are discussed (section 6).

General comments

The results part is well presented based on solid experiments (over 2 winters), with results that confirm the validity of the retrieval algorithm, showing a global relative uncertainty of about 11% compared to manual measurements and other sensors (Snow pillow and Snow scale). These results highlight the problem of certain assumptions used in the inversion (on density for example).
Have you looked at the ice crust effect (melting/freezing, or after a rain-on-snow event) in the snow?

It was foreseeable that the system would not be very efficient for monitoring precipitation events over a short period of time, given that the GNSS signal is integrated over 12 hours of measurements. This is a weak point of the system: 59% of events (Delta SWE>10 mm) was detected, see Table 3, but the exercise is interesting.

For the part of possible improvements of the system, it is clear that the current algorithm needs improvements, which are relatively little discussed in detail, but the authors argue that this was not the purpose of the article. OK

Regarding the improvement of snow height estimation, it is likely that adding GNSS signal analysis by reflectometry would improve the inversion process: but why was this not been done on the SnowSense? Are other specific antennas needed? More expensive? Longer processing time? Please specify.

I thus suggest "minor" correction with suggested clarifications.

Specific comments

- I suggest to use the term GNSS receiver (GNSSr) to name the snow measurement system based on GNSS signals.
- Introduction: I suggest to cite the recent review of SWE sensor (in review process, but probably published soon):
  Royer A., A. Roy, S. Jutras and A. Langlois (2021). Review article: Performance assessment of radiation-based field sensors for monitoring the water equivalent of snow cover (SWE). The Cryosphere Discuss. [preprint], https://doi.org/10.5194/tc-2021-163, in review, 2021.
- In the whole article, it is rather an uncertainty that is evaluated than an accuracy, since manual or other references also have their own, sometimes significant, uncertainties.
  For example, manual SWE measurement is subject to large variations and uncertainties, as studied in the revised version of Royer et al 's paper.
  Also, the Denoth system for measuring LWC can have large uncertainties (see the comparison paper:
  Mavrovic* A., J.-B. Madore*, A. Langlois, A. Royer and A. Roy (2020). Snow liquid water content measurement using an open-ended coaxial probe (OECP). Cold Regions Science and Technology. 171, 102958. )
- What do the red vertical bars in Figure 6 correspond to, for the manual LWC measurements?
- L121 The given speed of signal propagation in dry snow depends upon the density!
- L139 and 141 : what would be the impact in the retrieval of the these assumed limits: ($Ro\_s,dry,max$ and $Ro\_s,0$ ) ?
- L200 Define the acronym LTE
- Table 1 : precise the meaning of height of new snow (HN) and water equivalent of snowfall (HNW).
- L401 The results of this paper for the retrieved wet-snow SWE appears significantly better than those previously presented by Koch et al. (2019) ?
- Figure C1: Very interesting results! I might have put this figure in the results section! How did you differentiate between liquid and solid precipitations. The link between the amount of rain-on-snow and LWC would be original.

---

## Referee Comment (RC2)

**Review comments** on tc-2021-235 manuscript, entitled," GNSS signal-based snow water equivalent determination for different snowpack conditions along a steep elevation gradient".

**General comments**:

The tc-2021-235 manuscript, entitled," GNSS signal-based snow water equivalent determination for different snowpack conditions along a steep elevation gradient" presents the applicability of the GNSS-based SWE measurement at four different locations (820, 1185, 1510 and 2540 m a.s.l.) in the eastern Swiss Alps during two winter seasons (2018-2020). The aim of the study is to assess the performance of the GNSS algorithm which was described by Koch et al. (2019) and validated at the high-alpine site Weissfluhjoch (2540 m a.s.l) in the conditions of shallow snowpack, more frequent changes between dry- and wet-snow conditions.

As general comment, it is not evident that there are new conclusions on GNSS signal-based snow water equivalent determination rather than published ones as it is not clear whether different snow conditions studied newly; performance of the GNSS algorithm at 2540 m a.s.l. known and no much data at 820 m a.s.l. and not clear difference in snow condition at 1185 and 1510 m a.s.l. There was very good work done and analyses and conclusions were driven. Manuscript should be re-organized with a new title.

**Specific Comments:**

**Comment-1**: It was said that the performance of the GNSS algorithm was validated at the high-alpine site high-alpine site Weissfluhjoch (2540 m a.s.l) (Koch et al., 2019). And this site was again listed as one of 4 sites to be assessed in this study. This makes a bit confusing. Were the same results used from Koch et., 2019 or new analysis were done and added to this study? This should be explained well, and necessary modifications should be done.

**Comment-2**: In Line 205: It is said "We have chosen a 12-hour measurement period as it provides the best trade-off between accuracy and latency." It would be good to explain this selection with a bit more justification if possible. As the GNSS algorithm was validated in higher -alpine site where less changes in snow conditions. Were there any data / graphs related frequent changes between dry- and wet snow conditions at four test sites selected? Linking to this, in figure 3, color bar given in time when dry-snow and wet-snow GNSS algorithm were used. How were these algorithms decided to use?

**Comment-3**: The aim of the study was said to assess the performance of the GNSS algorithm in the conditions of shallow snowpack, more frequent changes between dry- and wet-snow conditions. Looking at the figure 3, and again color bar where dry-snow and wet-snow GNSS algorithm used, it looks to me there were no frequent changes in snow conditions or? It would be necessary to give information on snow conditions at test locations.

**Comment-4**: In Line 251: it is said that "For Küblis, only a qualitative evaluation was possible as there was hardly any snow during winter 2019-2020 and a long data gap in winter 2018-2019." Küblis 820 m a.s.l. (KUB) is important test location as lowest alpine-site where the conditions of shallow snowpack, more frequent changes between dry- and wet-snow conditions are higher probability as the purpose of the study. The Weissfluhjoch (2540 m a.s.l) site was already validated (Koch et al., 2019).

**Comment-5**: In Line 589 in Conclusion: It is said that "Overall, our analysis confirmed that the GNSS system can reliably measure the seasonal evolution of SWE at different elevations where different snow conditions prevail. We conclude that the GNSS-based derivation of SWE is a valuable, affordable and reliable alternative to manual measurements or other automated SWE sensors; the method is in principle suited for operational SWE monitoring. Moreover, the GNSS method represents to the best of our knowledge the most appropriate and cost-effective approach for measuring SWE and LWC simultaneously, continuously and non-destructively."

In this conclusion, what is new? There was already published studies in high-alpine snow conditions and there were no good data in lowest alpine, Küblis 820 m a.s.l. (KUB). Previous studies have also shown that the GNSS-based derivation of SWE is a valuable, affordable and reliable alternative to manual measurements or other automated SWE sensors; the method is in principle suited for operational SWE monitoring and etc.

---

## Author Comment (AC1)

**Reply to Referee #1: Alain Royer**

This paper completes the evaluation of the GNSS-derived approach for SWE monitoring using the retrieval algorithm already presented by Koch et al. (2019) and validated at the high-alpine site Weissfluhjoch (2540 m a.s.l), with data at 4 altitudes in the Alps (820, 1185, 1510 and 2540 m a.s.l.). The performance of this approach is thus assessed for shallow to deep snowpack, with more frequent changes between dry- and wet-snow conditions at low altitude, potential differences in densification and a higher influence of rain events compared to the high-alpine site Weissfluhjoch (2540 m a.s.l).

This article first presents the uncertainty results for the Snow Water Equivalent (SWE), the Liquide Water Content (LWC) and the snow depth (HS) estimates derived from SWE and LWC retrieved data for each of the 4 study sites. The authors then analyzed the potential detection of snow variations over a short period of time (24 h and 72 h) by comparing these with reference precipitation data, and discussed the current limitations in retrieving new snow. Since the retrieval of HS estimates and LWC parameter are derived using a recursive process from previously retrieved data, the authors assess also the stability of GNSS-derived snow parameters regarding data gaps. Lastly, outlook on potential improvements are discussed (section 6).

**General comments**

The results part is well presented based on solid experiments (over 2 winters), with results that confirm the validity of the retrieval algorithm, showing a global relative uncertainty of about 11% compared to manual measurements and other sensors (Snow pillow and Snow scale). These results highlight the problem of certain assumptions used in the inversion (on density for example). Have you looked at the ice crust effect (melting/freezing, or after a rain-on-snow event) in the snow?

Manually observed snow profiles were performed weekly for the sites of Laret, Klosters and Küblis and bi-weekly at Weissfluhjoch. Although we observed some ice layers and occasionally in spring the formation of a thick melt-freeze crust, we could not attribute any particular effects on the derived SWE to these ice layers based on the limited data. We assume that the effects of individual ice lenses are marginal on the measured bulk SWE as the dielectric property of ice are much more similar to the one of snow compared to water in wet snow.
Regarding the rain-on-snow events, we occasionally had rain at the Küblis and Klosters sites. However, these rain-on-snow events were too rare and did not allow for a detailed analysis. Therefore, we cannot make any firm statements on their influence on the GNSS-derived SWE or HS as we showed in the graph in Appendix C. Still, we can exclude that the rain-on-snow events had any major effects on SWE and HS as those would be visible in the graphs in Appendix C. In this regard, more research on rain-on snow events may be needed – we will add a sentence on this limitation in the Discussion section 5.1.

It was foreseeable that the system would not be very efficient for monitoring precipitation events over a short period of time, given that the GNSS signal is integrated over 12 hours of measurements. This is a weak point of the system: 59% of events (Delta SWE>10 mm) was detected, see Table 3, but the exercise is interesting.

For the part of possible improvements of the system, it is clear that the current algorithm needs improvements, which are relatively little discussed in detail, but the authors argue that this was not the purpose of the article. OK

Regarding the improvement of snow height estimation, it is likely that adding GNSS signal analysis by reflectometry would improve the inversion process: but why was this not been done on the SnowSense? Are other specific antennas needed? More expensive? Longer processing time? Please specify.

This study was particularly designed and executed to check if the SWE algorithm already tested and validated at the high-alpine site Weissfluhjoch works as well at lower laying alpine sites using the same hardware and algorithm setup. In the past, reflectometry has been applied mainly with geodetic antennas. Only recently it has been shown that reflectometry works with low-cost sensors as well. However, implementing reflectometry within our current algorithm would require a considerable adaptation, proper testing, and validation, which was beyond the scope of this study. As mentioned in the paper, we plan for including reflectometry in upcoming studies. It would allow us to evaluate whether it is possible to use reflectometry-derived snow depth instead of the dry/wet snow density models we currently use.

I thus suggest "minor" correction with suggested clarifications.

**Specific comments**
- I suggest to use the term GNSS receiver (GNSSr) to name the snow measurement system based on GNSS signals.

To be in line with our previous study we would like to keep the term we used. The term receiver has already been used for the electronic component. In fact, the entire measurement system is more than the receiver as it includes also e.g. the antenna and the processing board.

- Introduction: I suggest to cite the recent review of SWE sensor (in review process, but probably published soon): Royer A., A. Roy, S. Jutras and A. Langlois (2021). Review article: Performance assessment of radiation-based field sensors for monitoring the water equivalent of snow cover (SWE). The Cryosphere Discuss. [preprint], https://doi.org/10.5194/tc-2021-163, in review, 2021.

Thanks for pointing out this new publication. We will certainly cite it in the Introduction section – and hope you will reference our article as well.

- In the whole article, it is rather an uncertainty that is evaluated than an accuracy, since manual or other references also have their own, sometimes significant, uncertainties.
For example, manual SWE measurement is subject to large variations and uncertainties, as studied in the revised version of Royer et al 's paper. Also, the Denoth system for measuring LWC can have large uncertainties (see the comparison paper: Mavrovic* A., J.-B. Madore*, A. Langlois, A. Royer and A. Roy (2020). Snow liquid water content measurement using an open-ended coaxial probe (OECP). Cold Regions Science and Technology. 171, 102958. )

We agree and accordingly will substitute accuracy with uncertainty throughout the manuscript where appropriate. We point out in both Results and Discussion sections that the reference data for LWC are subjected to a large uncertainty in both text and figure (see next answer). We will add the reference mentioned.

- What do the red vertical bars in Figure 6 correspond to, for the manual LWC measurements?

The vertical bars indicate the estimated error. We will specify this in the figure caption.

- L121: The given speed of signal propagation in dry snow depends upon the density!

That's correct and we will point out in the revised manuscript that we used a mean value according to Schmid et al. (2014).

- L139 and 141: what would be the impact in the retrieval of the these assumed limits: (Ro_s,dry,max and Ro_s,0 ) ?

We do not think that discussing the impact of different model parameters is indicated at this point of the manuscript. These parameters were chosen to give best results for data at the location Weissfluhjoch (see Schmid et al., 2014). Changing these parameters will impact mainly the derived HS. As it was described in the Discussion (section 5.3) even a relatively large error in estimating HS has only a very small impact on the derivation of SWE.

- L200 Define the acronym LTE
We will refer to a mobile network data communication (LTE) module.

- Table 1 : precise the meaning of height of new snow (HN) and water equivalent of snowfall (HNW).

We will add in the figure caption that HN stands for height of new snow and HNW for water equivalent of snowfall.

- L401 The results of this paper for the retrieved wet-snow SWE appears significantly better than those previously presented by Koch et al. (2019) ?

The RMSE values of this study and Koch et al. (2019) cannot be compared directly. In this study, we compared the GNSS-derived SWE to manual SWE measurements. Whereas, as we explained in the following sentences (l. 401-404), Koch et al. 2019 compared the GNSS values with the automatic measurements from snow pillow and scale that exhibited significant errors at the beginning of the melt season. Therefore, the RMSE between the GNSS-derived SWE and the SWE measured by scale/pillow for wet-snow conditions is higher in Koch et al. (2019). We decided to base our comparison solely on the more trustful manual measurements because similar errors for snow pillow/scale were observed also in 2018-2019 and 2019-2020 (see Figure 3); moreover, for the sites Klosters and Küblis no scale or pillow measurements were available. We suppose that the accuracy was not lower in the former study – the difference is mainly related to the choice of reference data.

- Figure C1: Very interesting results! I might have put this figure in the results section! How did you differentiate between liquid and solid precipitations. The link between the amount of rain-on-snow and LWC would be original.

Indeed, it would be a very interesting link. However, at current stage, we think we have too few data to present significant results on this aspect. We hope to get more data including rain-on-snow events in the future to elaborate more on this. Regarding the discrimination between liquid and solid precipitation: as described in the figure caption, the precipitation was classified as snow below an air temperature threshold of T=1.1 °C and as rain above T=1.1 °C.

---

## Author Comment (AC2)

**Reply to Referee #2: A.N. Arslan**

**General comments:**

The tc-2021-235 manuscript, entitled," GNSS signal-based snow water equivalent determination for different snowpack conditions along a steep elevation gradient" presents the applicability of the GNSS-based SWE measurement at four different locations (820, 1185, 1510 and 2540 m a.s.l.) in the eastern Swiss Alps during two winter seasons (2018-2020). The aim of the study is to assess the performance of the GNSS algorithm which was described by Koch et al. (2019) and validated at the high-alpine site Weissfluhjoch (2540 m a.s.l) in the conditions of shallow snowpack, more frequent changes between dry- and wet-snow conditions.
As general comment, it is not evident that there are new conclusions on GNSS signal-based snow water equivalent determination rather than published ones as it is not clear whether different snow conditions studied newly; performance of the GNSS algorithm at 2540 m a.s.l. known and no much data at 820 m a.s.l. and not clear difference in snow condition at 1185 and 1510 m a.s.l. There was very good work done and analyses and conclusions were driven. Manuscript should be re-organized with a new title.

We regret that the novelty and differences to previous studies did not become clear. We will be happy to explain them once again.
Koch et al. 2019, tested the algorithm in detail at the high-alpine test site Weissfluhjoch, but it is still important to test and validate the method in lower laying regions with a shallower snowpack and more frequent changes from dry-snow to wet snow-conditions and vice versa as clearly mentioned in the manuscript. At the test site Weissfluhjoch, just one change from dry-snow to wet snow-conditions occurs in spring every year. The GNSS algorithm worked well with one seasonal change so far. However, at lower laying sites, the GNSS algorithm must change more frequently between different snow conditions. For example, by changing frequently between using the dry-snow density and the wet-snow density assumption.
Moreover, the snow conditions in Küblis, Klosters and Laret present some key differences: In Küblis, we only had a very shallow snowpack with predominantly wet-snow conditions including times with complete snow melt and small new snow events (season 2019-2020). Klosters, e.g., had more rain-on-snow events than the other stations. The total SWE varied also largely between the sites, e.g. for the season 2019-2020, the SWE in Laret was almost double the SWE in Klosters, but still approx. half of the SWE at the site Weissfluhjoch. In this context, it has to be mentioned that for a shallower snowpack, the portion of snow that is subject to daily melt-freeze cycles is larger and, therefore, the effect of the daily cycle is also larger.
We will add a sentence on this aspect in the Discussion section. In addition, we will explain the differences in snow conditions at the different elevations in more detail. We will also reformulate parts of the Introduction section and add a figure that illustrates the temperature evolution at the different sites and figures showing the rain on snow events for all sites.

Otherwise, we do not see any benefit in re-organizing the manuscript. We find the title and manuscript structure appropriate for the presented research question, results and conclusions.

**Specific Comments:**

Comment-1: It was said that the performance of the GNSS algorithm was validated at the high-alpine site high-alpine site Weissfluhjoch (2540 m a.s.l) (Koch et al., 2019). And this site was again listed as one of 4 sites to be assessed in this study. This makes a bit confusing. Were the same results used from Koch et., 2019 or new analysis were done and added to this study? This should be explained well, and necessary modifications should be done.

Actually, the time periods which were used in the current study and those used by Koch et al. (2019) are well defined in the papers. Koch et al. (2019) used data from the seasons 2015-2016, 2016-2017, and 2017-2018. The present study is based solely on data from the seasons 2018-2019 and 2019-2020.
In the present study, on one hand, we confirm the performance reported in Koch et al. (2019), on the other hand, more importantly, we compare the performance at the Weissfluhjoch site with the lower elevation sites; by doing so we can evaluate the performance for a much broader set of snow conditions.
We will emphasize more clearly that in the present study the data analyzed at the station Weissfluhjoch were solely from the seasons 2018-2019 and 2019-2020 .

Comment-2: In Line 205: It is said "We have chosen a 12-hour measurement period as it provides the best trade-off between accuracy and latency." It would be good to explain this selection with a bit more justification if possible. As the GNSS algorithm was validated in higher -alpine site where less changes in snow conditions. Were there any data / graphs related frequent changes between dry- and wet snow conditions at four test sites selected? Linking to this, in figure 3, color bar given in time when dry-snow and wet-snow GNSS algorithm were used. How were these algorithms decided to use?

The current algorithm works sufficiently well, when the time window of satellite observation and data collection is more than approx. 6 hours as it is necessary to capture as many satellites as possible with both, ascending and decreasing tracks. One satellite pass is on average approx. 6 hours. We choose a larger 12-hour time window (whenever possible, e.g., season 2019-2020) to improve the accuracy with respect to a 6-hour period, but there is actually no difference in performance, if the time span would be increased to one sidereal day (which represents the

entire coverage for the available GPS satellite before its repetition on the following sidereal day). We will add this information in Section 3.

The decision between dry- and wet snow-conditions is based on signal strength information as explained in Section 2 and Figure 2. We now included this information also in the caption of Figure 3. In addition, we also refer to Koch et al. (2019) for more detailed information on the algorithm itself and this particular decision step. We think this information should be sufficient to the reader. We also agree that changes between wet- and dry-snow conditions or vice versa can happen on a time scale shorter than 12 hours and will not be captured with the chosen data evaluation rate. Those rapid changes effect normally just the upper part of the snowpack and are relevant for the bulk properties only for a shallow snowpack. In general, the snow is classified as wet if a portion of the snowpack is wet for a short time span because the mean signal strength decreases sufficiently. In fact, we mentioned this issue in line 426 following: "The frequency of data sampling of 12 h used in this study did not allow to reveal the sub-daily wetting and refreezing cycle. However, LWC derivation at (half-)hourly frequency is possible and allows detecting sub-daily melt-freeze cycles as demonstrated by Koch et al. (2014) and Schmid et al. (2015)."

Comment-3: The aim of the study was said to assess the performance of the GNSS algorithm in the conditions of shallow snowpack, more frequent changes between dry- and wet-snow conditions. Looking at the figure 3, and again color bar where dry-snow and wet-snow GNSS algorithm used, it looks to me there were no frequent changes in snow conditions or? It would be necessary to give information on snow conditions at test locations.

Actually, there is a clear difference between the high-alpine site of Weissfluhjoch and the lower sites, which can also be seen in Figure 3, for example. At the high elevation site, the winter season is clearly divided in a dry accumulation and a melt season. For the lower sites it is different and melt (at least at the surface) occurs more frequently. The color bars in Figure 3 refer to the ''bulk'' state of the snowpack and do not reflect changes in only some parts of the snowpack. In fact, regarding daily melt-freeze cycles, for a shallower snowpack, the upper part has a large impact on the bulk value compared to a deeper snowpack where the effect is minor. We will add additional plots in the Appendices showing the frequency of rain-on-snow events and the temperature conditions for the four sites.

Comment-4: In Line 251: it is said that "For Küblis, only a qualitative evaluation was possible as there was hardly any snow during winter 2019-2020 and a long data gap in winter 2018-2019." Küblis 820 m a.s.l. (KUB) is important test location as lowest alpine-site where the conditions of

shallow snowpack, more frequent changes between dry- and wet-snow conditions are higher probability as the purpose of the study. The Weissfluhjoch (2540 m a.s.l) site was already validated (Koch et al., 2019).

We agree that it would have been desirable to have more data for the Küblis site, but unfortunately due to pole tilting in 2018-2019 and almost no snowfall in 2019-2020, those are the only data we have. However, as shown in Figure 3 and Appendix C, also at the stations Klosters and Laret more frequent changes and rain-on-snow events were observed. We think it is important to compare the performance at the Weissfluhjoch site with the lower elevation sites and to have a broader set of snow conditions in our analysis. Even though, we have few data at the Küblis site, we reached our aim of evaluation the method at sites where frequent changes in snow conditions and rain-on-snow events occur.

Comment-5: In Line 589 in Conclusion: It is said that "Overall, our analysis confirmed that the GNSS system can reliably measure the seasonal evolution of SWE at different elevations where different snow conditions prevail. We conclude that the GNSS-based derivation of SWE is a valuable, affordable and reliable alternative to manual measurements or other automated SWE sensors; the method is in principle suited for operational SWE monitoring. Moreover, the GNSS method represents to the best of our knowledge the most appropriate and cost-effective approach for measuring SWE and LWC simultaneously, continuously and non-destructively."

In this conclusion, what is new? There was already published studies in high-alpine snow conditions and there were no good data in lowest al-pine, Küblis 820 m a.s.l. (KUB). Previous studies have also shown that the GNSS-based derivation of SWE is a valuable, affordable and reliable alternative to manual measurements or other automated SWE sensors; the method is in principle suited for operational SWE monitoring and etc.

As outlined above, the novelty is that we tested the method for low elevation stations where frequent changes in snow conditions and rain-on-snow events occur. We are pleased that the reviewer shares our conclusion that the method is suited. Only with extensive testing, however, it is feasible to operationally implement a new method, in particular in the context of high-quality, long-term data series that are of high climatological relevance.

---

## Author Response (AR2)

**Answers to the Feedback by Reviewer #2 on the revised manuscript**

*Reviewer*: Thanks for updated manuscript. I believe that the manuscript has been improved. I have still following comments which I hope that updates will make positive impact on the paper.

*Authors*: We thank the reviewer, Dr. Arslan, for reviewing the updated version of the manuscript. Thanks also for the additional suggestions to further improve our manuscript – please see our answers below.

Comment 1:

*Reviewer*: Is measurement data (in-situ) available publicly so that may be readers can be interested on them? Would be good to give links where the data is available!

*Authors*: As already mentioned in the 'Data availability' section at the end of the manuscript, we uploaded the data shown in the paper to the environmental data portal and repository EnviDat. The data can be found at https://doi.org/10.16904/envidat.1869.

In addition, we now also added the in-situ manual profiles to the repository. Moreover, in response to your second comment, we also uploaded some webcam images.

Comment 2:

*Reviewer*: There has been mentioned that camera images are available and stored. It would be good to give some images where different snowpack conditions at different elevations are visible. This can be also as Annex. There has been some manuscript published at the Cryosphere related retrieval of snowpack live snow cover and snow depth with webcam images and also comparison done with manual and other automatic measurement sensors recently. It would be good to comment on those emergence technologies like webcams and drones on the retrieval of snowpack parameters.

*Authors*:  We now added to the publicly available data on the Envidat repository webcam images showing some typical differences in the snow conditions at the 4 sites for two time periods (14 to 18 March 2019 and 27 January to 11 February 2020). Both time periods include a precipitation event with partial rain-on-snow occurring at the lower locations.
Due to the relatively low quality of the images and the large amount of data, we find it excessive to add all pictures to the Envidat database. However, we mention in the file description that further webcam images can be made available to people interested in doing further analysis.

We agree with the reviewer that in general, webcams and drones are capable to increase our knowledge on snowpack properties. However, with webcam images, we cannot derive the GNSS-based bulk snow cover properties (SWE, HS, LWC), and therefore think there is no need to describe these optical approaches in the manuscript. In the Introduction section, we already mention numerous novel technologies including also references on drone- and satellite-based snow height derivations (please refer to lines 39-44).

Comment 3:

*Reviewer*: There has been work on the retrieval of snow depth using satellites like Sentinel-1. This will enable to estimate snow water equivalent in higher resolution. I believe it would be good to make some comments on that especially in discussions and conclusion.

*Authors*: We agree, and in fact already do mention these recent developments in Sentinal-1 based HS retrieval in the Introduction section. In the revised manuscript, we now added two additional references:

- Lievenset al. , 2021. Sentinel-1 snow depth retrieval at sub-kilometer resolution over the European Alps. The Cryosphere Discussions, pp.1-25.
- Tsang et al., 2021 Review Article: Global Monitoring of Snow Water Equivalent using High Frequency Radar Remote Sensing, The Cryosphere Discuss., 2021, 1-57).

Following your suggestion, we now also mention the novel approaches in satellite-based retrieval of SWE in the Discussion, in section 6.2, lines 570-572.

Comment 4:

*Reviewer*: I also think that it would be very good to have a comparison/summary table where different methods/measurement versus different snowpack parameters / conditions are visible. This will highlight the position of GNSS-based retrievals on different snowpack parameters and at different snowpack conditions.

*Authors*: We considered your suggestion, but we think that the relevant data and statistics are already shown in the manuscript quite extensively. We already show the detailed comparison of GNSS-derived SWE and HS and reference measurements in Table 2 and 4. For LWC, however, the accuracy of the reference data is quite low, and data are obviously available just for wet-snow conditions. For the specific condition 'rain-on-snow events' the available data are not sufficient to calculate proper statists as already mentioned. As also mentioned in the manuscript, we would like to collect more data, especially during rain-on-snow events, to come up with solid statistics for such conditions.